DESY-25-012

# How to Unfold Top Decays

Luigi Favaro[1,2], Roman Kogler[3], Alexander Paasch[4],
Sofia Palacios Schweitzer[1], Tilman Plehn[1,5] and Dennis Schwarz[6]

**1** Institut für Theoretische Physik, Universität Heidelberg, Germany
**2** CP3, Université catholique de Louvain, Louvain-la-Neuve, Belgium
**3** Deutsches Elektronen-Synchrotron DESY, Germany
**4** Institut für Experimentalphysik, Universität Hamburg, Germany
**5** Interdisciplinary Center for Scientific Computing (IWR), Universität Heidelberg, Germany
**6** Institute for High Energy Physics, Austrian Academy of Sciences, Austria

July 9, 2025

# Abstract

**Using unfolded top-quark decay data we can measure the top quark mass, as well as search for unexpected kinematic effects. We present a new generative unfolding method for the two tasks and show how they both benefit from unbinned, high-dimensional unfolding. Unlike weight-based or iterative generative methods we include a targeted unbiasing with respect to the training data. This shows significant advantages over standard, iterative methods, in terms of applicability, flexibility and accuracy.**

# 1 Introduction

Particle physics studies the fundamental properties of particles and their interactions, with the goal to discover physics beyond the Standard Model. The methodology is defined by the interplay between precision predictions and precision measurements. A key challenge is that perturbative quantum field theory makes predictions for partons, while experiments observe particles through their detector signatures. First-principle simulations link these two regimes [1]. They start with predictions for the hard process from a Lagrangian, and then add parton decays, QCD radiation, hadronization, and the detector response, to eventually compare with experimental data. This forward-simulation inference is the basis of, essentially, all LHC analyses.

The first problem with forward inference is that it requires access to the data and the entire simulation chain; neither of them are available outside the experimental collaborations. Second, it is not guaranteed that the best theory predictions are implemented in the forward simulation chain. Finally, in view of the high-luminosity LHC, hypothesis-driven forward analyses will overwhelm our computing resources for precision theory predictions and detector simulations. All three problems motivate alternative analysis techniques.

An exciting alternative analysis method is based on inverse simulations or unfolding. Instead of simulating detector effects for each predicted event, we can correct the observed events, for example, for detector effects. Then, we perform inference on particles before the detector or even partons and their hard scattering. Because the forward simulations are based on quantum physics and are stochastic, unfolding poses an incomplete inverse problem on a statistical basis. Still, in this way

1. analyses can be done outside the experimental collaborations;
2. theory predictions can be updated and improved easily;
3. and BSM hypotheses can be tested without full simulations.

Machine learning (ML) methods are revolutionizing not only our daily lives, but also LHC physics [2]. While classical unfolding methods are severely limited in many ways, ML-unfolding allows us to unfold unbinned events in many dimensions [3]. A reweighting-based ML-based unfolding method is MultiFold or OmniFold [4], applied to H1 [5–7], LHCb [8] and, recently, ATLAS [9] data. Generative ML-unfolding either maps distributions [10–14] or learns the underlying conditional probabilities [15–22]. Which of these complementary methods one would want to use depends on the specific task. Learning conditional probabilities to invert the forward simulation chain gives us access to per-event probabilities smoothly over phase space [23], guaranteeing the correct event migration. Its success rests on sufficiently precise generative networks [24–27], which are developed and benchmarked also for fast forward simulations [28–32]. In this paper we present a novel direction in ML-unfolding:

• we target an especially challenging task, mass measurement and unfolding of strongly peaked kinematics. Here, established methods, weight-based as well iterative generative unfolding, fail;

• we show the first unfolding results related to a CMS analysis [33, 34]. While this paper shows fast simulation results only, even more promising results for full CMS simulations can be obtained from the CMS members on our team.

This analysis also marks the first application of generative unfolding to properly simulated data by an LHC experiment. In Sec. 2 we describe the goal of the analysis, show the results from the classic CMS analysis, introduce the dataset, and sketch the basic features and the implementation of generative unfolding. In Sec. 3 we see how the top mass appears in the unfolded dataset. We find that a major problem is the uncontrolled bias induced by the training

data. It can be solved as described in Sec. 3.2. Next, we show in Sec. 3.3 how the top mass can be measured from the unfolded distributions, and in Sec. 3.4 we show how to then unfold the entire top decay phase space for re-analysis.

In App. A we illustrate how iterative bias removal methods do not work for peaked phase space distributions. The goal of this paper is to show that decay kinematics can be unfolded and to provide a blueprint for an LHC analysis using generative unfolding.

## 2   Goal and method

If we want to unfold top-quark decay events, the main challenge is the model dependence and resulting bias when the top masses assumed for the simulated training data and the actual top mass differ. We could attempt this with iterative improvements of the unfolding network [35], but we will see that this approach is numerically extremely challenging. We follow a slightly different strategy:

1. we ensure that the bias from the top mass assumed in the simulated training data is small;
2. we infer the correct top mass from the data, using a reduced unfolded phase space;
3. we produce training data with the inferred top mass and unfold the full phase space.

### 2.1   Top mass measurement

The extraction of the top mass from the invariant jet mass of highly boosted hadronic top quark decays can shed light on possible ambiguities in top mass measurements using simulated parton showers. The ultimate goal is to compare the measured jet mass distribution to predictions from analytic calculations. For that, it is convenient to unfold detector effects.

Unfolding uses simulated data, biasing the unfolded data towards the model used in the simulation. In particular, the choice of the top mass in the simulation leads to a significant uncertainty [34]. These modelling biases can be reduced by including more information and granularity into the unfolding process, motivating the use of ML-unfolding methods.

In the existing CMS measurement this is done by also unfolding differentially in the top-jet transverse momentum and by including various sideband regions close to the measurement phase space. Using ML-unfolding, the data can be unfolded in a larger number of phase space dimensions, providing ways to reduce the model bias.

The result from our CMS benchmark analysis [34] is shown in Fig. 1. This analysis unfolds the reconstructed 3-subjet mass $M_{jjj}$ and the corresponding reconstructed transverse momentum, $p_{T,jjj}$ to measure the top mass. The three subjets are obtained using a two-step clustering with the eXclusive Cone (XCone) algorithm [36]. In the first step, the event is clustered into two large-radius jets with a distance parameter $R = 1.2$ to capture the decay products of the top quark and antiquark. In a second step, the two large-radius jets from the first step are each reclustered into three XCone subjets with $R = 0.4$, where the subjets capture the dynamics of the hadronic top quark decay. Before the unfolding, the jet mass scale is calibrated by reconstructing the $W$-boson from the two light-quark subjets and fitting the subjet energy scales to the resulting $W$ peak. The $W$ boson decay is identified with the help of $b$-tagging information, which is obtained for the XCone subjets by matching these in angular distance to small-$R$ anti-$k_T$ jets. This matching is needed because the $b$-tagging information is not calculated for XCone subjets in CMS. The uncertainty in the unfolding from the modeling of final state radiation is reduced with the help of another auxiliary measurement of $N$-subjettiness ratios [37] on large-$R$ anti-$k_T$ jets, matched by angular closeness to the large-$R$ XCone jets. The matching

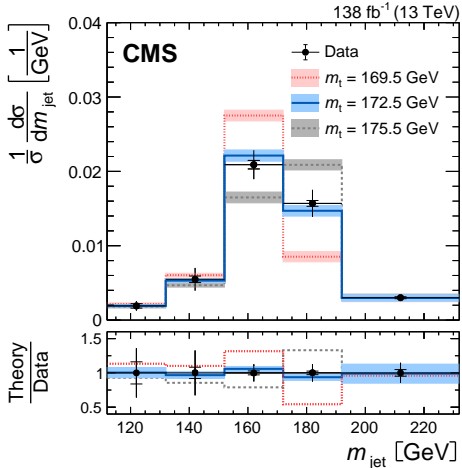

Figure 1: CMS benchmark result from Ref. [34]. It shows the differential top pair cross section as a function of the top-jet invariant mass, compared to theory predictions for different top masses. The vertical bars represent the total uncertainties, statistical uncertainties are shown as short horizontal bars, and theoretical uncertainties as shaded bands.

procedures and auxiliary measurements add considerable complications to the measurement and come with non-negligible uncertainties. Because of the finite efficiency of the $b$ tagging and the associated mis-identification rate, the information from the $W$ reconstruction cannot be used in the unfolding because it breaks the permutation invariance among the jets. The leading systematic uncertainties in this measurement originate from the jet energy scale, jet mass scale, jet mass resolution, the $b$-jet response and the unfolding bias from the choice of the top mass in the simulation. Non-negligible uncertainties also arise from the modeling of non-perturbative effects. Ideally, unfolding enough phase space dimensions to capture the $W$ decay and the salient features of the jet substructure should allow us to constrain the dominating uncertainties in-situ and remove the top-mass bias in the unfolding.

Once we have measured the jet mass in an event sample and consequently the top mass, we can further analyze the unfolded dataset. For instance, we can look for effects from higher-dimensional SMEFT operators on the decay of boosted tops, or we can search for anomalous kinematic distributions from new particles, modified interactions, or enhanced QCD effects at the subjet level. While the unfolding for the top mass measurement has to include a sufficiently large number of dimensions, as discussed above, we now need to unfold the full, 12-dimensional phase space. Three of these dimensions are finite jet masses, generated by QCD effects.

## 2.2 Dataset

We use simulated events for top pair production, similar to the one used for a CMS measurement [34]. We generate the events with Madgraph 5 [38]. Hadronization, parton showers, and multiple parton interactions are simulated with Pythia 8.230 [39] with the underlying event tune CP5 [40]. The samples include a simulation of the detector response implemented in Delphes 3.5.0 [41] using the default CMS card with pile-up, and the e-flow algorithm. The pile-up subtraction only removes charged tracks associated to pile-up vertices. This simulation is a Delphes version of the CMS simulation for Ref. [34].

In the simulated data, we have access to three stages of the simulation chain. The parton level includes the hard interactions of the top quarks, that decay into a $b$-quark and a $W$-boson,

that subsequently decays into two quarks or lepton and neutrino. The particle level refers to all stable particles with lifetimes longer than $10^{-8}$ s after parton shower and hadronization. Finally, the detector level describes particle candidates after the detector simulation. At this point, we limit ourselves to events which appear at all three stages, our results show that the treatment of efficiency effects is sub-leading and beyond the scope of this study.

Event selections are applied at the particle and detector level. All events that do not pass either of the selections are rejected from further analysis. For the signal or measurement region, we only consider $t\bar{t}$ pairs in the lepton+jets decay at the parton level,

$$pp \to t\bar{t} \to (bq\bar{q}')\,(\bar{b}\ell^-\bar{\nu}) + \text{c.c.} \quad \text{with} \quad \ell = e, \mu\,, \tag{1}$$

with the lepton acceptance

$$p_{T,\ell} > 60\,\text{GeV} \quad \text{and} \quad |\eta_\ell| < 2.4\,. \tag{2}$$

The top jet is constructed using XCone clustering and identified by the larger angular distance to the lepton. It must fulfill

$$p_{T,J} > 400\,\text{GeV} \qquad \text{and} \qquad p_{T,j_{1,2,3}} > 30\,\text{GeV} \qquad |\eta_{j_{1,2,3}}| < 2.5\,, \tag{3}$$

for the large-$R$ jet $J$ and three subjets $j_i$. In the following, we will refer to these subjets as jets. The second large-$R$ jet has to have $p_{T,J} > 10\,\text{GeV}$ to reject poorly reconstructed events where only the lepton and not the $b$ quark is reconstructed in the second large-$R$ jet. To reduce the contribution from events where the full top quark decay is not reconstructed within the top jet, we require the invariant mass of the three jets, $M_{jjj}$, to exceed the invariant mass of the lepton and the large-$R$ jet close to it.

At the detector level, in addition to the above requirements, the missing transverse momentum has to be larger than $50\,\text{GeV}$ and at least one $b$-tagged jet must be present.

The measurement-region selection criteria leave us with approximately 800,000 events simulated with a top mass of $m_t = 172.5\,\text{GeV}$, of which we use 75% for the training. To be consistent with the amount of events available in CMS with the full detector simulations for the reference analysis, we choose samples with different top masses to have less events. All events contain the full generator (gen) and reconstruction (reco) level information. The XCone algorithm clusters the jets separately for reco-level jets and gen-level jets. The clustered jets are sorted according to $p_T$.

We only consider paired events in our signal, i.e. events that passed both reco- and gen-level cuts. Non-paired events can be treated as background if they are selected at the reco-level but are not part of the measurement's fiducial phase space at the gen-level. On the other hand, events that were generated in the fiducial phase space at gen-level but were not reconstructed because of the detector's acceptance or an inefficiency will need to be accounted for by an efficiency correction. This can be done through weights, as for example done in the Iterative Bayesian unfolding method [42–45] as implemented in RooUnfold [46] and in TUnfold [47], and successfully applied in several jet substructure analyses at the LHC, see for example Refs. [34,48–50]. Another way to include efficiency and acceptance effects is through a classifier [51], but we leave the details of such a study to future work as these are closely related to the actual implementation of the data analysis.

The CMS analysis [34] shows that continuum backgrounds, like $W$+jets production, can be subtracted bin-wise to the level where they are no longer relevant for in the analysis. The normalization uncertainties in the different backgrounds introduce a shape uncertainty when changing the normalization of single processes. While the background normalizations vary between 20–100% in the CMS analysis, the overall background uncertainty was estimated to

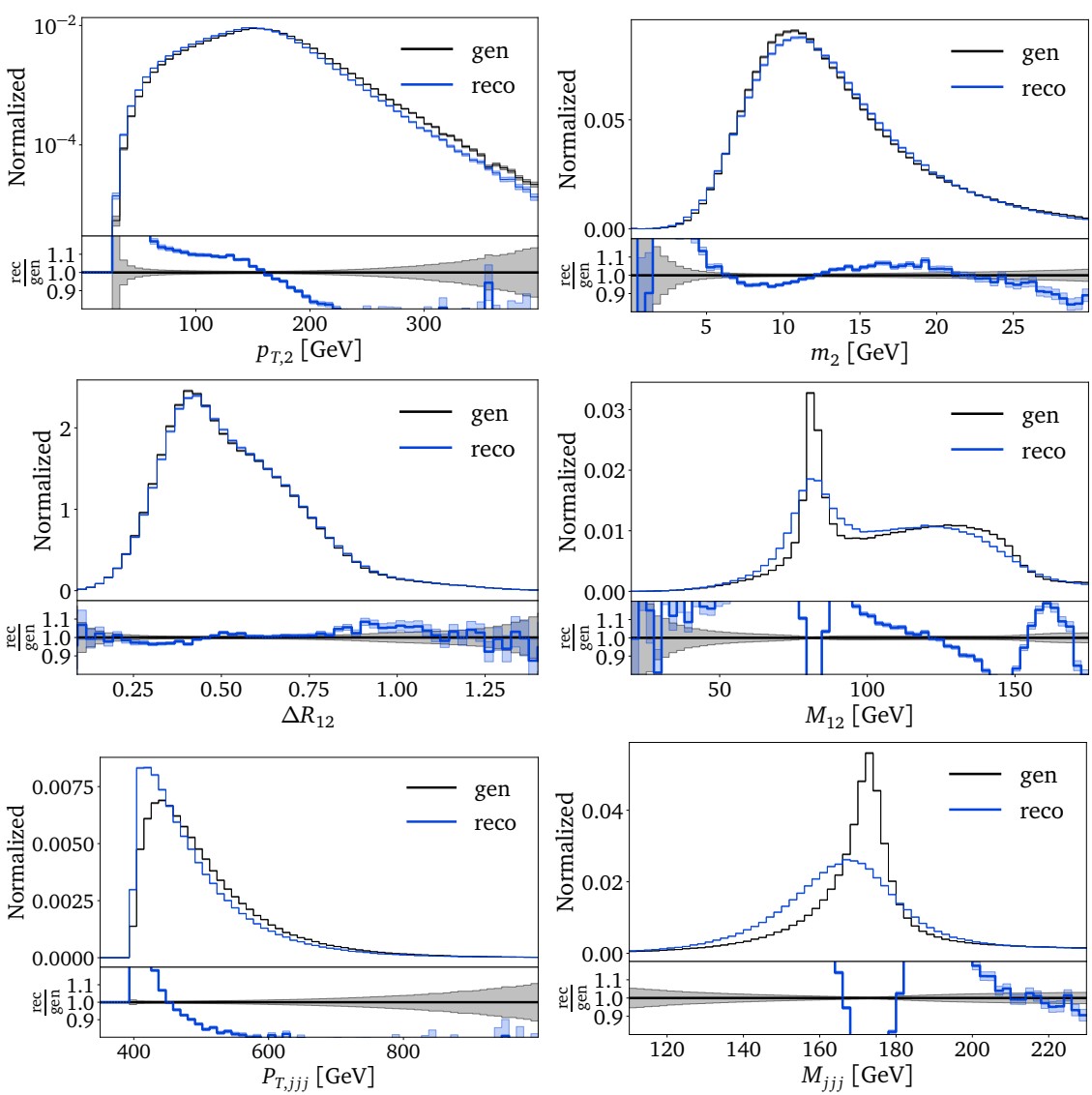

Figure 2: Kinematic distributions at reco-level and gen-level for the second jet (top), combining two jets (center), and combining three jets (bottom).

be only 0.01 GeV in the extraction of the top quark mass and is thus negligible compared to other uncertainties. The method of bin-wise background subtraction can be generalized to the unbinned case with the help of a classifier [52], which suggests that background uncertainties will remain small compared to other systematic uncertainties in this measurement. Therefore, we neglect these in our study and consider signal events only.

## 2.3 Jet-mass features

For the generative unfolding algorithm a perfect matching between reco-level and gen-level jets is not critical, as the reco-level is used only as a condition. We have checked that when permuting the ordering of the reco-level jets randomly, we observe no difference in performance. Once we switch to the 4-momentum representation $(m, p_T, \phi, \eta)$, we see small differences between reco-level and gen-level, for instance in the $p_T$ and individual jet masses shown in Fig. 2 (top row).

Differences in the jet masses are mostly due to pile-up in our simulation, which is added

at the reco-level, and to a lesser degree from inefficiencies and mis-reconstructions in the reconstruction of photons, charged and neutral hadrons. Pile-up contributions are reduced by removing tracks originating from pile-up vertices. The remaining difference in the jet mass mostly comes from photons and neutral hadrons in the pile-up.This positive contribution to the jet masses is largest for the leading jet because of its larger $p_T$ compared to the other jets. Figure 2 implies that unfolding detector effects includes correcting for these pile-up effects. As Delphes assumes an idealized vertex reconstruction, we expect those differences to be larger when including full detector effects with GEANT4 [53].

Going beyond single-jet observables, we need to understand and eventually unfold detector effects on jet-jet correlations. In Fig. 2 (middle row) we show two examples. The distribution in the angular separation between the two leading jets shows a characteristic peak, originating from the boosted decay kinematics combined with mass effects and the detector acceptance. The 2-jet masses have a peculiar distribution, owed to the fact that out of the three jets two come from the $W$ decay. Because of the $p_T$-ordering, any of the three combinations

$$M_{ik}^2 = m_i^2 + m_k^2 + 2 \left( m_{T,i} m_{T,k} \cosh \Delta y_{ik} - p_{T,i} p_{T,k} \cos \Delta \phi_{ik} \right) \tag{4}$$

can reconstruct $m_W$. This is an exact equation for the three 2-jet masses, where $\Delta y_{ik}$ represents the difference in jet rapidities. Of the three 2-jet masses in a top decay, two tend to be similarly close to $M_{ik} \sim m_W$ [54]. In Fig. 2 (middle right), we also observe the upper endpoint in the top decay kinematics at gen-level [55]

$$m_{bj}^{\max} < \sqrt{m_t^2 - m_W^2} \approx 155 \, \text{GeV} \,. \tag{5}$$

Following Eq.(4), we can improve the training of the unfolding network by including the 2-jet masses as explicit features. Each of the 2-jet masses then substitutes an angular variable. With this basis transformation we sacrifice access to the individual azimuthal angles and are left with their absolute differences.

Next, we see in Fig. 2 (bottom row) that the transverse top quark momentum is not affected significantly by detector effects, and the 3-jet mass peaks around the top mass value. In our phase space parametrization we can calculate the 3-jet mass as

$$M_{jjj}^2 = M_{12}^2 + M_{23}^2 + M_{13}^2 - m_1^2 - m_2^2 - m_3^2. \tag{6}$$

By using all these jet masses as training features, we can greatly improve the learning and unfolding of the 3-jet mass. The no-free-lunch theorem, however, tells us that this gain will lead to a mismodelling of other correlations. In particular, we will see that there is no guarantee that $\cos \Delta \phi \in [0, 1]$ anymore, leading to the generation of unphysical event kinematics in some cases.

## 2.4  Generative unfolding

Traditional unfolding algorithms [56–58] have been used to unfold simple differential cross section measurements. Widely used methods include Iterative Bayesian Unfolding [42–45], Singular Value Decomposition [59], and TUnfold [47]. Their limitation is the need for binned data in a low-dimensional phase space. This also means that we have to preselect the observables we want to unfold and decide on their binning before the unfolding.

To use ML-methods for high-dimensional and unbinned unfolding, we invert the forward simulation using Bayes' theorem

$$p(x_{\text{gen}}|x_{\text{reco}}) = p(x_{\text{reco}}|x_{\text{gen}}) \frac{w(x_{\text{gen}})p(x_{\text{gen}})}{w(x_{\text{reco}})p(x_{\text{reco}})} \,, \tag{7}$$

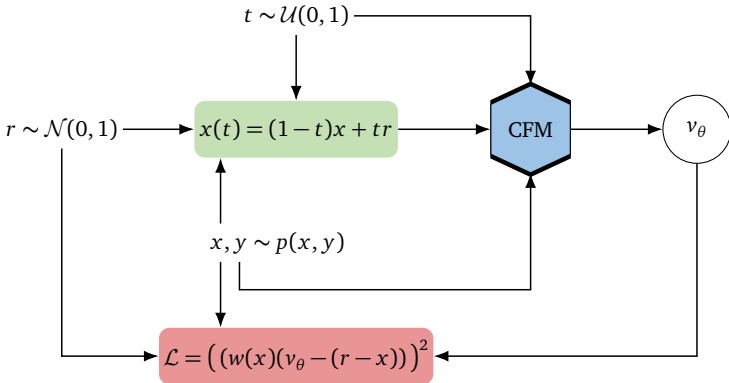

Figure 3: Schematic representation of generative unfolding with a CFM network.

where $x_{\text{gen}}$ is a point in the weighted gen-level phase space and $x_{\text{reco}}$ a point in the weighted reco-level phase space. The gen-level and reco-level weights are encoded by $w(x_{\text{gen}})$ and $w(x_{\text{reco}})$ respectively. To unfold reco-level data, we need to learn

$$p_{\text{model}}(x_{\text{gen}}|x_{\text{reco}}) \approx p(x_{\text{gen}}|x_{\text{reco}}) \tag{8}$$

as the statistical basis of an inverse simulation. The network endoding this conditional probability can be a GAN [15], an INN version of a normalizing flow [16], or a diffusion network [3]. Once a generative neural network encodes $p_{\text{model}}(x_{\text{gen}}|x_{\text{reco}})$, we calculate

$$p_{\text{unfold}}(x_{\text{gen}}) = \int dx_{\text{reco}}\, p_{\text{model}}(x_{\text{gen}}|x_{\text{reco}})w(x_{\text{reco}})p(x_{\text{reco}})\,. \tag{9}$$

At the event level, this integral can easily be evaluated by marginalizing the corresponding joint probability. Our method can be summarized as

$$
\begin{array}{ccc}
p_{\text{sim}}(x_{\text{gen}}) & & p_{\text{unfold}}(x_{\text{gen}}) \\
\text{paired data} \Big\uparrow & & \Big\uparrow {\scriptstyle p_{\text{model}}(x_{\text{gen}}|x_{\text{reco}})} \\
p_{\text{sim}}(x_{\text{reco}}) & \xleftrightarrow{\text{correspondence}} & p_{\text{data}}(x_{\text{reco}})\,.
\end{array} \tag{10}
$$

The two distributions $p_{\text{sim}}(x_{\text{reco}})$ and $p_{\text{sim}}(x_{\text{gen}})$ are encoded in one set of simulated events, before and after detector effects, or at the parton- and the reco-level.

The generative network we employ to learn $p_{\text{model}}(x_{\text{gen}}|x_{\text{reco}})$ is Conditional Flow Matching (CFM). The generative CFM network is the leading architecture for precision-LHC simulations [26]. Mathematically, CFM is based on two equivalent ways of describing a diffusion process using an ordinary differential equation (ODE) or a continuity equation [60]

$$\frac{dx(t)}{dt} = v(x(t), t) \qquad \text{or} \qquad \frac{\partial p(x, t)}{\partial t} = -\nabla_\theta \left[ v(x(t), t)p(x(t), t) \right], \tag{11}$$

both with the same velocity field $v(x(t), t)$. The diffusion process described by $t \in [0, 1]$ relates a latent Gaussian distribution $p_{\text{latent}}(r)$ to the physical phase space $p_{\text{data}}(x)$,

$$p(x, t) \to \begin{cases} p_{\text{data}}(x) & t \to 0 \\ p_{\text{latent}}(r) = \mathcal{N}(r; 0, 1) & t \to 1\,. \end{cases} \tag{12}$$

254 We employ a simple linear interpolation

$$x(t) = (1-t)x + tr \rightarrow \begin{cases} x & t \rightarrow 0 \\ r \sim \mathcal{N}(0,1) & t \rightarrow 1 \, . \end{cases} \tag{13}$$

255 Using this approximation, we train the network to learn

$$v_\theta(x(t), t) \approx v(x(t), t) \tag{14}$$

256 using the continuity equation and then generate phase space configurations using a fast ODE
257 solver. Even though the corresponding MSE loss function

$$\mathcal{L}_{\text{CFM}} = [w(x)(v_\theta - (r - x))]^2 \tag{15}$$

258 is not a likelihood loss, a Bayesian version of the CFM generative network can learn uncer-
259 tainties on the underlying phase space density together with the central values underlying its
260 sampling [26].

261     The CFM setup is illustrated in Fig. 3. Its conditional extension is straightforward, in
262 complete analogy to the conditional GANs [15] and conditional INNs [16] developed for un-
263 folding. While the naive GAN setup does not learn the event-wise (inverse) migration correctly
264 and therefore does not encode physical, calibrated conditional probabilities, the cINN with its
265 likelihood loss does exactly that. The CFM succeeds because of its mathematical foundation,
266 Eq.(11) [3].

## Training bias

268 In Eq.(10) we describe the structure of generative unfolding, but we are missing a critical
269 complication — the simulated reco-level data $p_{\text{sim}}(x_{\text{reco}})$ might not agree with the actual reco-
270 level data $p_{\text{data}}(x_{\text{reco}})$.

271     Let us assume a simple case where the simulation depends on a simulation parameter $m_s$
272 which we can tune to describe the actual data. This can be a physics parameter we eventually
273 infer, or a nuisance parameter which we profile over. The dependencies of the four datasets
274 on $m_s$ and its 'correct' value in the data, $m_d$, turn Eq.(10) into

$$\begin{array}{ccc} p_{\text{sim}}(x_{\text{gen}}|m_s) & & p_{\text{unfold}}(x_{\text{gen}}|m_s, m_d) \\ \scriptstyle p(x_{\text{reco}}|x_{\text{gen}}) \Big\downarrow & & \Big\uparrow \scriptstyle p_{\text{model}}(x_{\text{gen}}|x_{\text{reco}}, m_s) \\ p_{\text{sim}}(x_{\text{reco}}|m_s) & \xrightarrow{\text{correspondence}} & p_{\text{data}}(x_{\text{reco}}|m_d) \, . \end{array} \tag{16}$$

275 In the forward direction, $p(x_{\text{reco}}|x_{\text{gen}})$ does not have an explicit $m_s$-dependence, but both
276 simulated datasets follow $p_{\text{sim}}(x_{\text{gen}}|m_s)$ and $p_{\text{sim}}(x_{\text{reco}}|m_s)$ induced by the generator settings.
277 By assumption, $m_s = m_d$ ensures that the simulated and actual data agree at the reco-level,

$$p_{\text{sim}}(x_{\text{reco}}|m_s = m_d) \overset{!}{=} p_{\text{data}}(x_{\text{reco}}|m_d) \, . \tag{17}$$

278 We then use this relation to infer $m_d$ at the reco-level.

279     Alternatively, we can do the same inference at the gen-level, requiring

$$p_{\text{sim}}(x_{\text{gen}}|m_s = m_d) \overset{!}{=} p_{\text{unfold}}(x_{\text{gen}}|m_s = m_d, m_d) \, . \tag{18}$$

The problem with this unfolded inference is the dual dependence of $p_{\mathrm{unfold}}(x_{\mathrm{gen}}|m_s, m_d)$ through the reco-level data and the learned conditional probability. This dual dependence is automatically resolved if $p_{\mathrm{unfold}}(x_{\mathrm{gen}})$ only depends on $m_d$ through the reco-level data, so the bias from $p_{\mathrm{model}}(x_{\mathrm{gen}}|x_{\mathrm{reco}}, m_s)$ can be neglected. It is important to emphasize that such a bias from the training data would lead to an uncontrolled systematic shift and a wrongly measured mass value.

An established way to remove the bias is through iteratively re- weighting the training dataset. This IcINN method [35] can of course applied to any conditional generative network. It relies on a learned classifier over $x_{\mathrm{gen}}$ which reweights $p_{\mathrm{sim}}$ to $p_{\mathrm{unfold}}$ including the $m_s$-dependencies and serves as a basis for re-training the unfolding network. It implicitly assumes that $p_{\mathrm{unfold}}(x_{\mathrm{gen}}|m_s, m_d)$ depends mostly on $m_d$ and at a reduced level on $m_s$. In that case the endpoint of the Bayesian iteration is reached when the two dependencies coincide at the level of the remaining statistical uncertainty. In App. A we show results for top decays and discuss the reasons for them not working.

# 3 Unbinned top quark decay unfolding

Unfolding top decays is technically challenging, because the top mass and the $W$ mass are dominant features of an altogether 12-dimensional phase space. We start with a naive unfolding in Sec. 3.1, using our appropriate phase space parametrization with reduced dimensionality [20]. In Sec. 3.2, we show how the model dependence from the top mass in the training data can be controlled. With this enhancement, we show in Sec. 3.3 how the high-dimensional unfolding improves the existing top mass measurement based on classic unfolding. Finally, we show how to unfold the entire 12-dimensional phase space using the measured top mass in Sec. 3.4.

## 3.1 Lower-dimensional unfolding

We know that the precision of learned phase space distribution using neural networks scales unfavorably with the phase space dimension [61, 62].* The full 12-dimensional phase space will not be the optimal representation to measure the top mass. Instead, we only use a lower-dimensional phase space representation for the top mass measurement, finding a balance between relevant kinematic information and dimensionality. We postpone the full kinematic unfolding to the point where we need to access the full kinematics and benefit from the measured top mass.

For the traditional CMS analysis [34], two phase space dimensions were unfolded, $M_{jjj}$ and $p_{T,jjj}$, where the $p_{T,jjj}$ was integrated over in the final measurement. The jet mass calibration relies on the reconstructed $W$ boson. Identifying the $W$-decay jets in the top jet ideally requires $b$-tagging information, but because of the inefficiency not all jets from the $W$ decay can be identified. Instead, the jet mass can be calibrated by using all possible 2-jet combinations, where each of the three resulting distributions feature a sharp $W$-mass peak (see Fig. 2). Therefore, we unfold those for the top mass measurement such that a reliable calibration can be performed at a later stage.

Our unfolding setup follows Sec. 2.3. From Eq.(6) we know that we can extract the 3-jet mass as a proxy for the top mass from the set of single-jet and 2-jet masses. Because the single-jet masses are largely universal and not a good handle on the jet energy calibration, our first

---

*For a possible improvement see Ref. [63, 64].

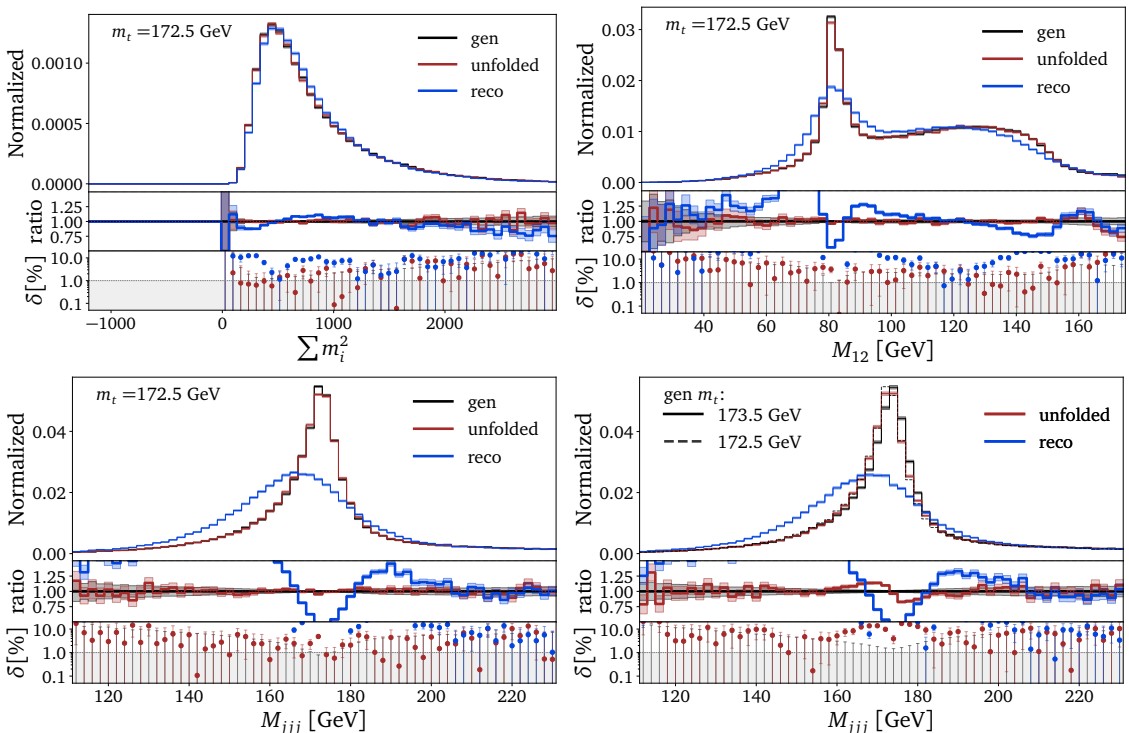

Figure 4: Kinematic distributions from the 4-dimensional unfolding. We also show the reco-level and the gen-level truth for $m_t = 172.5\,\text{GeV}$. In the bottom-right panel we compare $M_{jjj}$ for $m_t = 172.5\,\text{GeV}$ to generated unfolding for $m_t = 173.5\,\text{GeV}$, not seen during training.

choice is to measure the top mass from a 4-dimensional unfolding of

$$\left\{ M_{j1j2}, M_{j2j3}, M_{j1j3}, \sum_i m_i \right\}. \tag{19}$$

The results are shown in Fig. 4. First, we see that we can unfold the sum of the single jet masses extremely well, with deviations of the unfolded data from the generator truth at the per-cent level. This means that we expect to be able to extract the 3-jet mass essentially from the sum of all 2-jet masses with a known and controlled offset.

Next, we show a 2-jet mass, with the characteristic $W$ peak and the shoulder at $m_{bj}^{\text{max}}$. The $W$ peak is washed out at the reco-level, but the generative unfolding reproduces the gen-level extremely well. The relative deviation of the unfolded to the truth 2-jet mass distributions is at most a few per-cent, with no visible shift around the $W$ peak. The same quality of the unfolding can be observed in the $M_{jjj}$ distribution, perfectly reproducing the top mass at $m_t = 172.5\,\text{GeV}$, the correct value in the training data and in the data which gets unfolded.

The problem with measuring the top mass from unfolded data appears when we unfold data simulated with a different top mass. In the lower-right panel of Fig. 4 we show the unfolded $M_{jjj}$ distribution for reco-level data generated with $m_t = 173.5\,\text{GeV}$, unfolded with generative networks trained on $m_t = 172.5\,\text{GeV}$. We see that the top peak in the unfolded data is dominated by the training bias of the network, specifically a maximum at $M_{jjj} = (172 \pm 1)\,\text{GeV}$. This means the top peak is entirely determined by the training bias and hardly impacted by the reco-level data which we unfold.

From the 4-dimensional unfolding we know that the network learns the $W$ peak in the 2-jet masses and the top peak in the 3-jet mass at a precision much below the physical particle widths. The problem is that the bias from the network training completely determines the

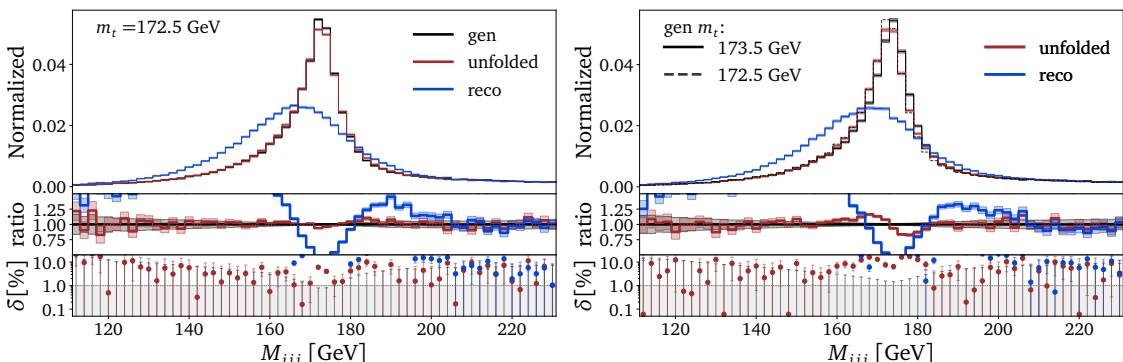

Figure 5: Kinematic distributions from 6-dimensional unfolding. In the right panel we compare $M_{jjj}$ for $m_t = 172.5\,\text{GeV}$ to generated unfolding for $m_t = 173.5\,\text{GeV}$, not seen during training.

position of these mass peaks in the unfolded data. To confirm that these findings are not an artifact of our reduced phase space dimensionality, we repeat the same analysis for the 6-dimensional phase space

$$\left\{ M_{j_1 j_2}, M_{j_2 j_3}, M_{j_1 j_3}, m_{j1}, m_{j2}, m_{j3} \right\}. \tag{20}$$

The unfolded 3-jet mass distributions are shown in Fig. 5, corresponding to the 4-dimensional case in Fig. 4. While the unfolded peak in $M_{jjj}$ is a bit worse than for the easier 4-dimensional case when unfolding the same value of $m_t$ as used in the training, the bias from the training remains in spite of the fact that we are weakening the expressive power of the unfolding network by adding distributions that are mildly affected by the peak position.

Finally, it is instructive to study the true and learned migrations between the reco-level and the gen-level 3-jet distribution. These are shown in Fig. 6, where in the left panel we see that the forward simulation maps the sharp peak at gen-level to a broader peak at reco-level. The problem with the central ellipse describing this physical migration by detector effects is that it does not indicate any correlation between the $M_{jjj}$-values at reco-level and at gen-level. The learned migration in the right panel reproduces the forward migration exactly.

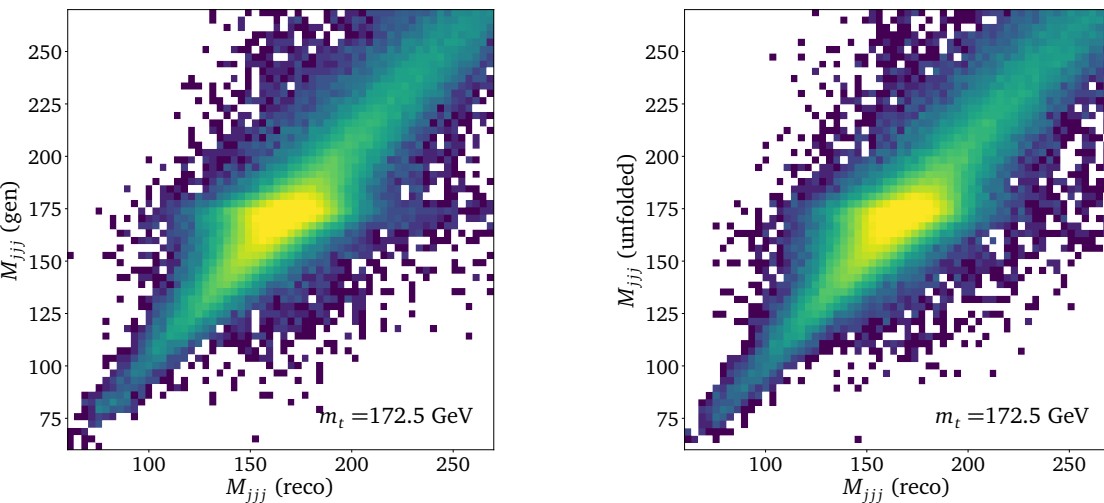

Figure 6: True and learned migrations in the $M_{jjj}$ distribution between reco-level and gen-level.

356    For the generative unfolding this means that small differences at reco-level will always be
357 unfolded to the same sharp region at gen-level, independent of the information contained in
358 the reco-level data. Following Sec. 2.4 and Eq.(16) the unfolded distribution $p_{\text{unfold}}(x_{\text{gen}})$ is
359 entirely determined by the training choice $m_s$ and shows practically no dependence on the
360 value $m_d$ encoded in the actual data.

## 3.2 Taming the training bias

362 The next question is how we can improve the situation where, $m_s$ being the top mass value
363 used for the simulation and $m_d$ the actual top mass in the data, Eq.(16) turns into

$$
\begin{array}{ccc}
p_{\text{sim}}(x_{\text{gen}}|m_s) & & p_{\text{unfold}}(x_{\text{gen}}|m_s, \cancel{m_d}) \\
\\
p(x_{\text{reco}}|x_{\text{gen}}) \Big\downarrow & & \Big\uparrow p_{\text{model}}(x_{\text{gen}}|x_{\text{reco}}, m_s) \\
\\
p_{\text{sim}}(x_{\text{reco}}|m_s) & \xleftrightarrow{\text{correspondence}} & p_{\text{data}}(x_{\text{reco}}|m_d).
\end{array}
\tag{21}
$$

364 In the unfolded distribution, the training information $m_s$ completely overwrites $m_d$. More-
365 over, even if there was enough sensitivity, a classifier comparing two shifted mass peaks learns
366 weights far away from unity, leading to numerical challenges. This means we cannot use the
367 usual iterative methods to remove the bias from the training data.

368    Following the strategy from Sec. 2, we first increase the sensitivity on $m_d$. For this, we
369 pre-process the data such that $m_d$ is directly accessible by adding an estimator of $m_d$ to the
370 representation of $x_{\text{reco}}$. Ideally, this estimator would be inspired by an optimal observable.
371 Such a one-dimensional observable with sufficient statistical precision should exist, and we
372 know how to construct it. For the top mass we just use the weighted median of the 3-jet masses
373 at reco-level, $M_{jjj}^{\text{batch}} = \frac{1}{N_{\text{batch}}} \sum_i^{N_{\text{batch}}} M_{jjj,i}$, where the sum runs over all, possibly weighted,
374 events in one batch. For a batch size around $10^4$ events, this information will be strongly
375 correlated with the top mass,

$$
M_{jjj}^{\text{batch}} \approx m_d \equiv m_t \Big|_{\text{data}}.
\tag{22}
$$

376 This batch-wise kinematic information can be extracted at the level of the loss evaluation, and it
377 goes beyond the usual single-event information, similar to established MMD loss modifications
378 of GAN training [15, 24].

379    Second, we weaken the bias from the training data by combining training data with dif-
380 ferent top masses, but without an additional label,

$$
m_t = \{169.5, 172.5, 175.5\} \text{ GeV} \qquad \text{(combined training)}.
\tag{23}
$$

381 It turns out that it is sufficient to cover a range of top masses with separate, unmixed training
382 batches. The range ensures that top masses in the actual data are within the range of the train-
383 ing data. We ensure a balanced training by enlarging the event samples with $m_t = 169.5$ and
384 175.5 GeV to match the size of the largest sample. This is done by repeating and shuffling the
385 input data, which effectively uses these events several times per epoch. we avoid overfitting
386 using an appropriate regularization. The limited number of simulated events for the eventual
387 analysis makes this training strategy sub-optimal. We expect larger and additional $m_t$ simula-
388 tions, unavailable at this time, to improve the results. As shown in App. A, both steps need to
389 be included to ensure precise, unbiased results.

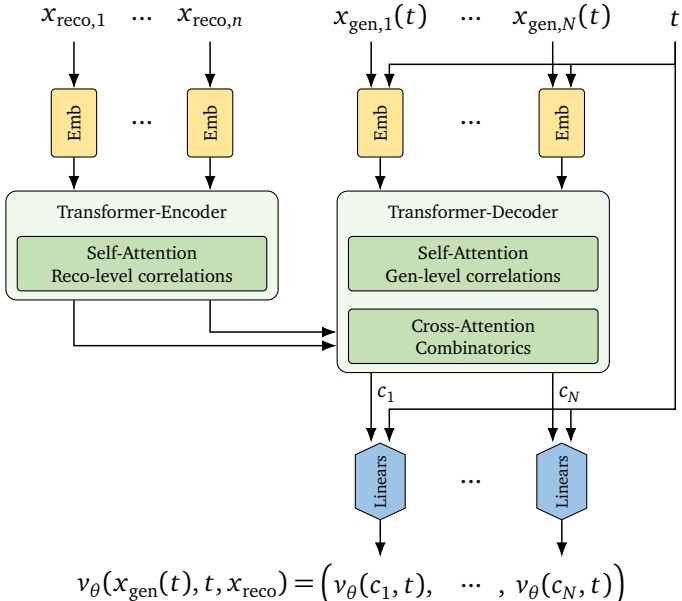

Figure 7: Schematic representation of a parallel transfusion network, adapted from [3].

Obviously, this strategy of strengthening the dependence on $m_d$ and reducing the training bias is not applicable to all problems, and it does not lead to the endpoint of the Bayesian iterative method, but for our combined inference-unfolding strategy it works, and this is all we need.

**Transfusion architecture**

As the network task becomes significantly more difficult we replace the simple dense architecture with a transfusion network, described in detail in Refs. [3, 51] and visualized in Fig. 7.

Each component of the $n$-dimensional condition as well as of the time-dependent $N$-dimensional input $x(t)$ are individually embedded by concatenating positional information and zero padding. The embedded conditions are passed through the encoder part of a transformer, while the embedded input is passed through the decoder counterpart. In both transformer parts, we apply self-attention to learn the correlations in the condition and in the input. The network is complemented by a cross-attention between encoder and decoder outputs, to learn the correlations between conditions and inputs. These are crucial for the unfolding task. For every component of the input, the transformer returns one high-dimensional embedding vector $c_i$, which is mapped back to a one-dimensional component of the velocity field by a shared dense linear network. This way, we express the learned $N$-dimensional velocity field of Eq.(14) as

$$v_\theta(x_{\text{gen}}(t), t, x_{\text{reco}}) = (v_\theta(c_1, t), \ldots, v_\theta(c_N, t)). \qquad (24)$$

The hyperparameters of the network can be found in Appendix B.

Using the transfusion network we unfold the 4-dimensional phase space from Eq.(19). The results are shown in Fig. 8 (top row). We unfold data generated with two different top masses, $m_t = 171.5$ and $173.5$ GeV. Neither of these two values are present in the training data. We observe in both cases that the top mass as the main kinematic feature is reproduced well, without a significant deviation from the gen-level distributions. The fitted peak values of the distributions are $m_{\text{peak}} = (172 \pm 1)$ GeV when unfolding data with $m_t = 171.5$ GeV, and

$m_{\text{peak}} = (174 \pm 1)\,\text{GeV}$ when unfolding data with $m_t = 173.5\,\text{GeV}$. While the bias might not have vanished entirely, it is well contained within the numerical uncertainties. We will extract the unfolded top mass value properly in Sec. 3.3.

**Dual network**

Given the more complicated training task, we observe a drop in performance when we increase the dimensionality to unfold the 6-dimensional phase space

$$x = \left( \{m_i\}, \{M_{ik}\} \right), \tag{25}$$

defined in Eq.(20) using the transfusion network. Inspired by Refs. [25, 26], we factorize the phase space density into two parts, each encoded in a generative network: the first network learns the individual jet mass directions in phase space, which are universal and do not depend on the value of $m_t$; the second network generates the 2-jet masses conditioned on the individual jet masses,

$$p(x_{\text{gen}}|x_{\text{reco}}) = \underbrace{p\left( \{m_{i,\text{gen}}\} \middle| x_{\text{reco}}, M_{jjj}^{\text{batch}} \right)}_{\text{network 1}} \underbrace{p\left( \{M_{ik,\text{gen}}\} \middle| \{m_{i,\text{gen}}\}, x_{\text{reco}}, M_{jjj}^{\text{batch}} \right)}_{\text{network 2}}. \tag{26}$$

Both CFM-transfusion networks also receive $M_{jjj}^{\text{batch}}$ calculated for a full batch using Eq.(6). For the event generation we first generate the unfolded jet masses $\{m_i\}$, pass them as a condition to the second network, and then generate the unfolded 2-jet masses $\{M_{ik}\}$.

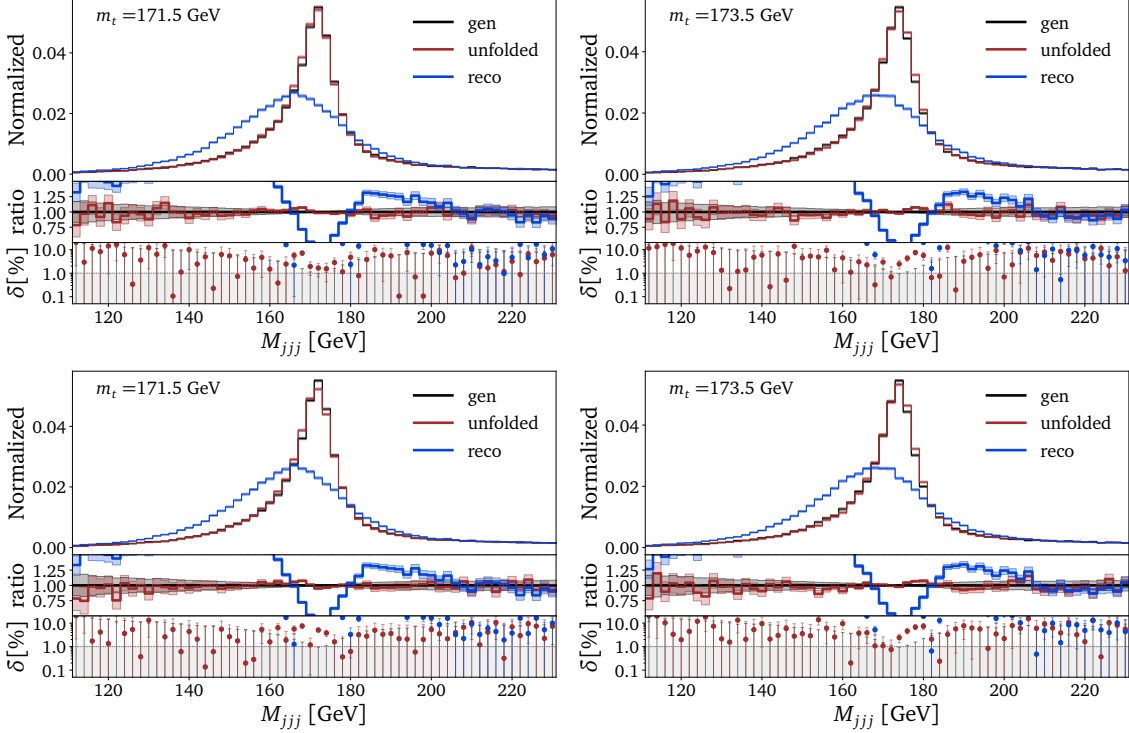

Figure 8: $M_{jjj}$-distributions from the 4-dimensional (top row) and 6-dimensional (bottom row) unfolding of data with $m_t = 171.5\,\text{GeV}$ (left column) and $m_t = 173.5\,\text{GeV}$ (right column). We train the network combining samples with three top masses, Eq.(23).

429     Looking at the 6-dimensional correlation giving $M_{jjj}$ in Fig. 8 (bottom row), we observe
430 a hardly visible drop in performance, but still no bias from the training data. As before, we
431 observe peak values at $m_{\text{peak}} = (172 \pm 1)\,$GeV when unfolding data with $m_t = 171.5\,$GeV and
432 at $m_{\text{peak}} = (174 \pm 1)\,$GeV when unfolding data with $m_t = 173.5\,$GeV.

### 3.3   Mock top quark mass measurement

434 We estimate the benefit from generative unfolding by repeating the top quark mass measure-
435 ment from Ref. [34], but with a large number of bins in the $M_{jjj}$ histogram. The top mass is
436 extracted from the binned unfolded distributions using a fit based on $\chi^2 = d^T V^{-1} d$, where
437 $d$ is the vector of bin-wise differences between the normalized unfolded distribution and the
438 normalized prediction from the simulated data. The covariance matrix $V$ contains the uncer-
439 tainties and corresponding bin-to-bin correlations. A parabola fit provides the central value of
440 $m_t$ and the standard deviation. Experimental systematics and simulation uncertainties have
441 to be propagated to the top mass measurements [34], combined with an in-situ jet calibration
442 using the known $W$-mass peak. Crucially, these uncertainties do not lead to an uncontrolled
443 bias of the unfolding, but will typically manifest themselves as noise.

**Statistical and model uncertainties**

445 First, this fit requires the covariance matrix describing statistical uncertainties [65]. We sample
446 $N$ times from the latent space, conditional on the reco-level events. This means we generate $N$
447 unfolded distributions from the posterior $p_{\text{model}}(x_{\text{gen}}|x_{\text{reco}})$. We then use a Poisson bootstrap,
448 where we assign a weight from a Poisson distribution with unit mean. The size of one replica
449 is 52,000 events, corresponding to the approximate number of real data events. The number
450 of events follows a Poisson distribution, with the mean given by the nominal sample size.

451     For the measurement, we create $N_{\text{rep}} = 1000$ replicas by selecting the nominal number of
452 reco-level events from the test dataset with $m_t = 172.5\,$GeV and the full datasets for the simu-
453 lations at different top masses. We unfold each replica, calculate $M_{jjj}$, and use the histogram
454 entries $u_i^{(n)}$ to compute the correlation matrix of statistical fluctuations as

$$\text{cov}_{ij} = \frac{1}{N_{\text{rep}}} \sum_{n=1}^{N_{\text{rep}}} (u_i^{(n)} - \bar{u}_i)(u_j^{(n)} - \bar{u}_j) \qquad \text{with} \qquad \bar{u}_i = \frac{1}{N_{\text{rep}}} \sum_{n=1}^{N_{\text{rep}}} u_i^{(n)}$$

$$\rho_{ij} = \frac{\text{cov}_{ij}}{\sqrt{\text{cov}_{ii}} \sqrt{\text{cov}_{jj}}} \ . \tag{27}$$

455 This procedure also takes into account the uncertainties due to the statistical fluctuations of
456 $M_{jjj}^{\text{batch}}$. The training of the network itself introduces correlations which are at least one order
457 of magnitude smaller and therefore ignored in the measurement.

458     The $5 \times 5$ and $60 \times 60$ correlation matrices $\rho_{ij}$ from the 4-dimensional unfolding using
459 the largest sample generated with $m_t = 172.5\,$GeV are shown in Fig. 9. We see two distinct
460 sources of bin-to-bin correlations. In general, an event migrating from bin $i$ to bin $j$ gives rise
461 to negative correlations in $\rho_{ij}$ between the two bins. Additionally, unbiasing the unfolding
462 ensures that a shift in the batch-wise condition also shifts the unfolded peak. This effect,
463 accounted for in the bootstrapping method, introduces an additional contribution to the bin-
464 to-bin correlations. It causes positive correlations between bins on the same side of the peak
465 and anti-correlations otherwise. In our case, both effects are most apparent in the peak region
466 and its neighboring bins.

467     We follow Ref. [34] to estimate the uncertainty from the choice of $m_t$ in the simulation used
468 for the unfolding. We evaluate the difference in each bin $i$ between the unfolded distribution

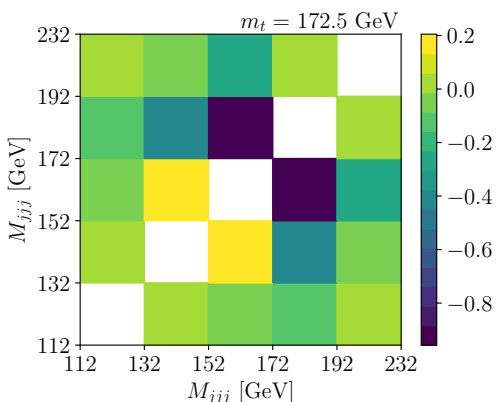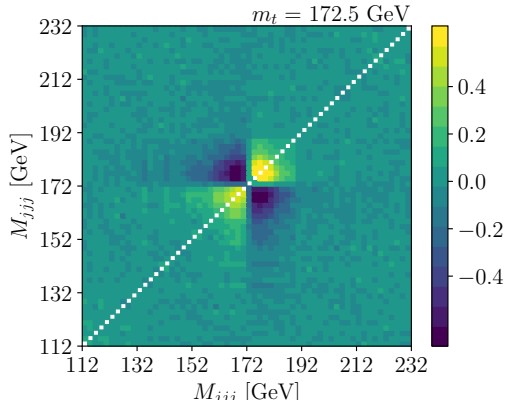

Figure 9: Correlation matrices obtained from $N_{\text{rep}} = 1000$ replicas for 5 bins (left) and 60 bins (right) in the 4-dimension unfolding with $m_t = 172.5$ GeV.

and the corresponding simulated gen-level distribution. From the differences $d_i$, we construct a covariance matrix

$$\text{cov}_{ij}^{\text{model}} = \rho_{ij} d_i d_j \,, \tag{28}$$

where $\rho_{ij}$ are the correlations between bins $i$ and $j$. Because the bin-to-bin correlations are not known and we do not observe any systematic pattern, we choose a diagonal covariance matrix with $\rho_{ij} = 1$ for $i = j$ and $\rho_{ij} = 0$ otherwise. It was verified that other choices do not alter the results. To estimate the impact of this model uncertainty, we perform the $m_t$ extraction twice. First, we only include the statistical covariance matrix corresponding to 52,000 available events at the reco-level. Second, we repeat the same measurement also including the model uncertainty.

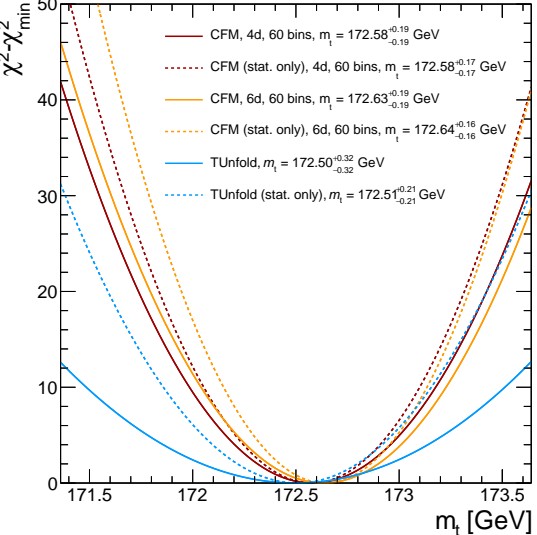

Figure 10: Extraction of $m_t$ with a $\chi^2$ test. The dotted lines include only statistical uncertainties, while the solid lines also include the model uncertainty from the choice of $m_t$.

**Improvement**

To compare our new unfolding technique to the existing TUnfold results [34], we repeat the extraction using the simulated data set with 172.5 GeV and using the statistical covariance matrix from the measured data, published in HEPData [66]. The $\chi^2$-curves and the corresponding results are displayed in Fig. 10, where we show the 4-dimensional and 6-dimensional unfoldings with 60 bins and the TUnfold result. We see that the uncertainty in the choice of $m_t$ is reduced from being a leading model uncertainty in the CMS measurement to a much smaller level. The statistical uncertainty in the TUnfold result was already small relative to the systematic uncertainties. Both the 4-dimensional and 6-dimensional unfoldings exhibit comparable statistical uncertainties with the 5-bin configuration. However, increasing the number of bins leads to a reduction in statistical uncertainty, as demonstrated below.

To confirm that the choice in $m_t$ does not leave a residual bias, we repeat the top quark mass extraction for unfolded data obtained from reco-level data simulated with different top masses. The results are shown in the left panel of Fig. 11. For a top mass of $m_t = 173.5$ GeV, we observe a bias of about 0.5 GeV when using a measurement with 5 bins. This is not surprising as the exact binning has been optimized for a minimal model dependence in the CMS measurement, which we did not do here. While the bin width in the unfolding with TUnfold is limited by the jet mass resolution, we test various binning schemes for the unbinned unfolding. The bias gets reduced when using more bins in the measurement, as expected because the binning introduces a regularization in the unfolding which leads to a model dependence. With 10 and more measurement bins, we observe that the bias from the model dependence is removed. For more measurement bins than 60, the comparably coarse grid of gen-level distributions with $m_t = \{169.5, 171.5, 172.5, 173.5, 175.5\}$ GeV leads to an unstable closure test.

To circumvent this limitation, we interpolate the gen-level distributions for $m_t$-values close to 172.5 GeV, where three samples with a separation of 1 GeV are available and a linear dependence of the bin content as a function of $m_t$ represents a valid approximation. Now, we can compare the resulting values of $m_t$ from the generative unfolding with 5 to 60 bins in terms of the statistical uncertainty. The results are displayed in the right panel of Fig. 11, indicating an increase in the statistical precision in $m_t$ due to the improved resolution.

## 3.4 Full phase space unfolding

As a last step of our unfolding program, we unfold the full 12-dimensional phase space given the measured top mass. This has the advantage that the leading source of training bias is removed. Following the same precision arguments as before, we keep the mass basis of Eq.(20) for the first 6 of the 12 phase space dimensions. This ensures that the 2-jet and 3-jet masses are reproduced well, albeit not at the level of the dedicated first unfolding step.

The remaining phase space dimensions are

$$x = \left( \{m_i\}, \{M_{ik}\}, \{p_{T,i}\}, \{\eta_i\} \right) \qquad i, k = 1, 2, 3 \,, \tag{29}$$

all other kinematic observables can be computed from these basis directions. For the 12-dimensional unfolding we use a single transfusion network, after checking that the dual network does not lead to an improvement. The hyperparameters are given in Appendix B. Two kinematic distributions are shown in Fig. 12. In the left panel, we see that the top mass peak is learned almost as well as for the 4-dimensional and 6-dimensional cases. Indeed, this is the case for all jet masses and 2-jets masses, which are combined to the 3-jet mass with the top peak.

A serious issue arises from the azimuthal angle between the two leading jets, $|\Delta\phi_{12}|$. According to Eq.(4) this angle is learned as a correlation of 7 phase space directions. Moreover,

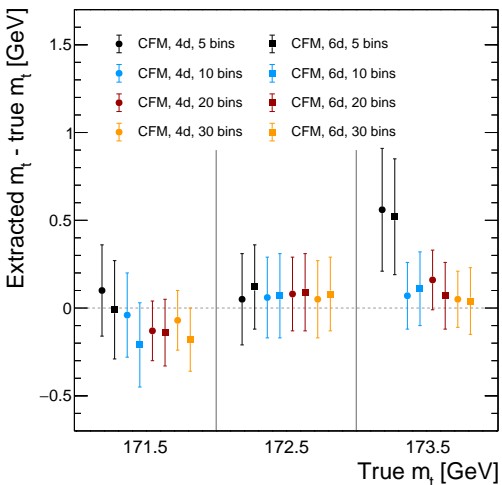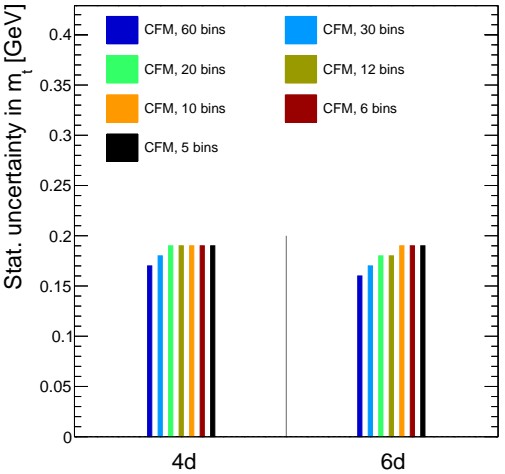

Figure 11: Deviation of the extracted top mass from the reco-level truth, employing 4-dimensional and 6-dimensional unfoldings with different numbers of measurement bins for $m_t = 171.5$, $172.5$, and $173.5$ GeV (left) . The size of the statistical uncertainties in $m_t$ from the 4-dimensional and 6-dimensional unfoldings with different binnings, assuming $m_t = 172.5$ GeV (right).

we do not have access to the azimuthal angles, only to the cosine of differences between angles. Here the problem arises that the network does not ensure that this cosine comes out in the physical range $-1 \ldots 1$. We enforce the physical range by clipping the cosine for small angles to one, which causes a mis-modelling of the small-$|\Delta\phi_{12}|$ regime, shown in the right panel of Fig. 13.

A simple way to improve this mis-modelling is to require $\cos\Delta\phi_{12} < 1$. However, from Fig. 12 we know that this does not solve the problem. Instead, we accept the fact that for unfolding the masses well we might have to pay a prize in the coverage of the angular correlations, and we apply an additional acceptance cut

$$\Delta\phi_{ik} > 0.1 \tag{30}$$

both, at the reco- and gen-levels in our simulated events. This reduces the size of the unfolded dataset by 30%. An extended set of unfolded kinematic distribution after this cut are shown in Fig. 13.

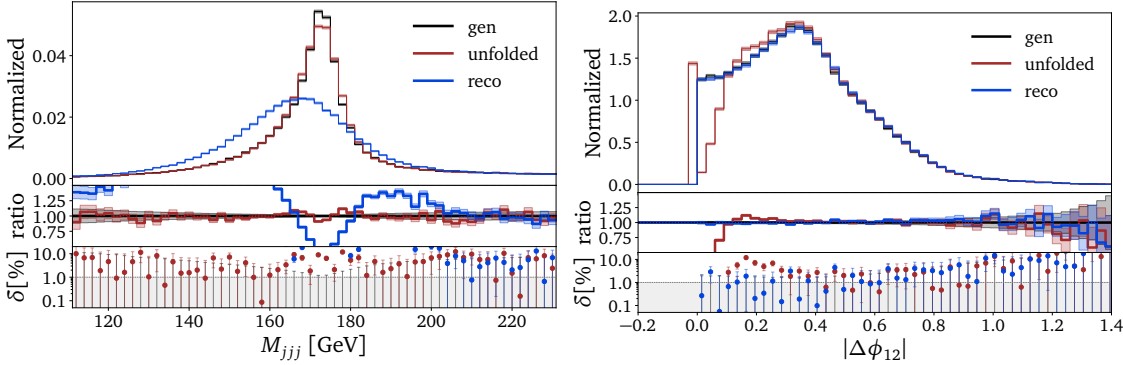

Figure 12: Kinematic distributions from full, 12-dimensional unfolding. We show the 3-jet mass as well as the azimuthal angle between the two leading jets.

535 We know that our unfolding method covers correlations between the original phase space
536 directions well, because many of the kinematic observables shown in Fig. 13 are built from
537 complex correlations of our phase space basis. However, to end with a nice figure and to
538 drive home the message that high-dimensional unfolding using conditional generative net-

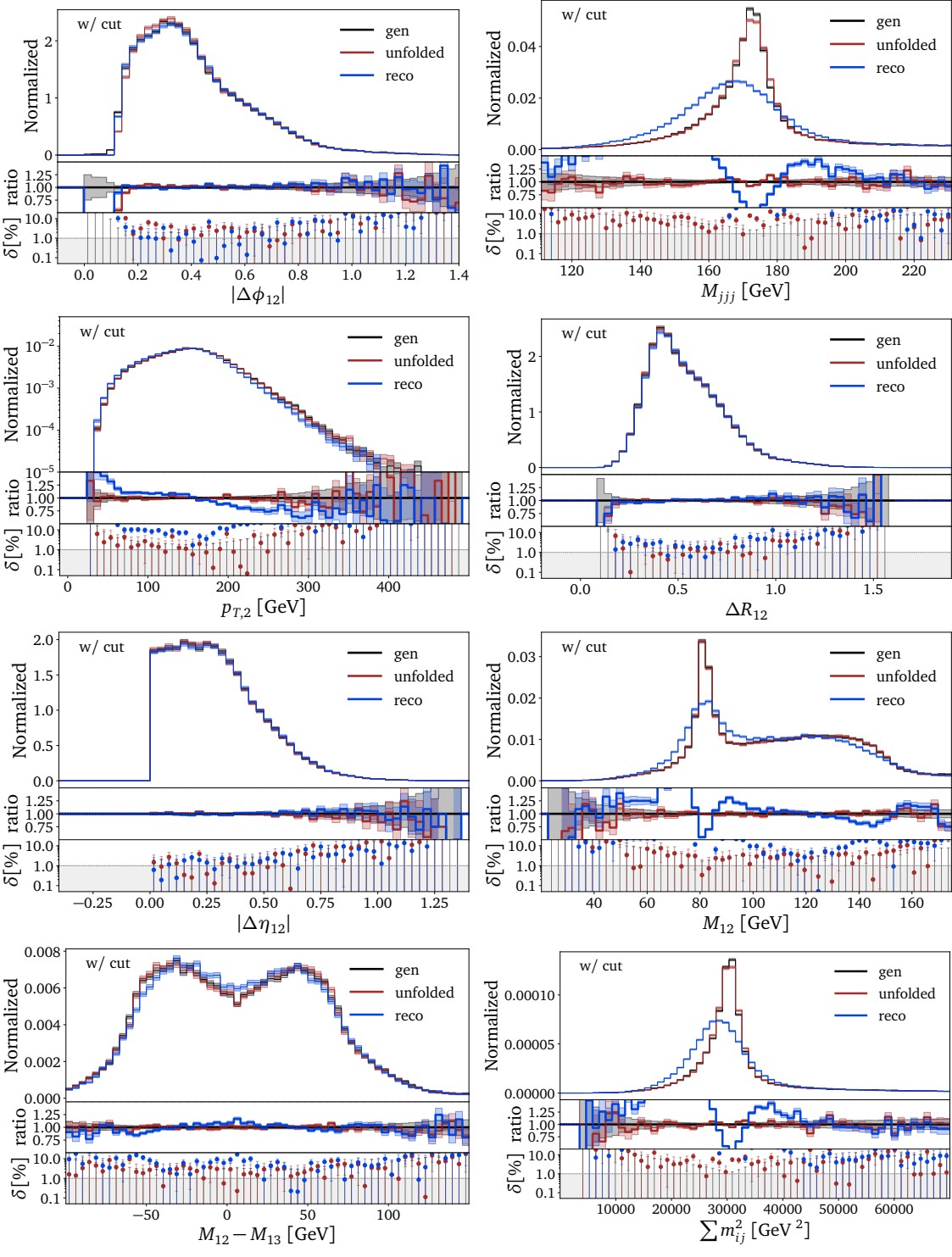

Figure 13: Kinematic distributions from full, 12-dimensional unfolding. We show the target
3-jet distribution, the azimuthal angle between the jets after cut, and a set of single-jet observ-
ables, 2-jet correlations, and 3-jet correlations (top to bottom).

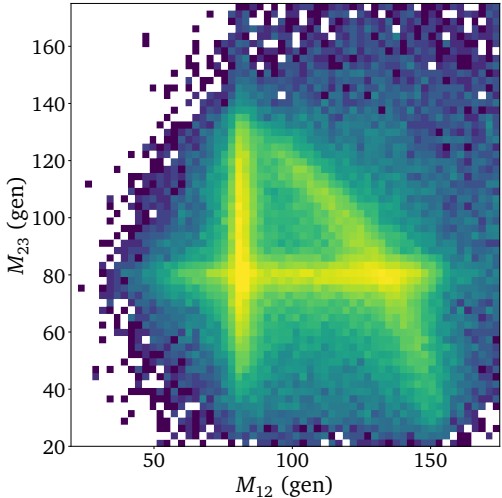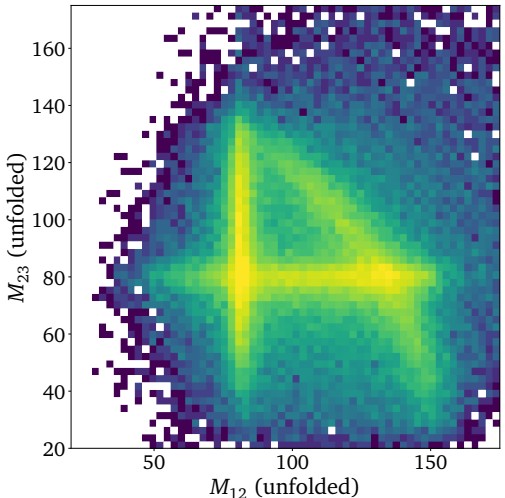

Figure 14: Correlation of two 2-jet masses at gen-level truth (left) and after unfolding (right).

works does learn the corresponding correlations well, we show one of our favorite correlations in Fig. 14. Indeed, there is literally no difference in the correlations between two of the three 2-jet masses. This correlation also confirms that the condition $M_{ik} \approx m_W$ leads to three distinct lines in phase space, where close to the crossing points it is impossible to reconstruct which two of the jets come from the $W$ decay.

# 4 Outlook

Unfolding is one of the ways modern machine learning is transforming the way we can do LHC physics. Employing an inverse simulation, it allows for the efficient analysis of LHC data by the LHC collaborations, to combine analyses between different experiments, and even make unbinned, unfolded data accessible to researchers outside the experimental collaborations. Unfolding has been used in particle physics frequently, but modern neural networks allow us to unfold a high-dimensional phase space without a choice of binning. This technical advance will turn multi-dimensional and unbinned unfolding into a standard analysis method at the LHC and future experiments.

For our study, we unfold detector effects from boosted top quark decay data using state-of-the-art conditional generative networks. Unfolding decay kinematics is especially challenging because we expect a large model dependence and even systematic bias from the choice of the top mass in the simulated training data. Our study shows that generative unfolding with a new methods for prior removal solves this problem and provides a first milestone towards incorporating generative unfolding in an existing CMS analysis.

First, we showed that for an appropriate phase space parametrization, a combination of diffusion network and transformer can reliably unfold a 4-dimensional and 6-dimensional subspace of the full top-decay phase space at the percent level precision. This included the 3-jet mass as a proxy to the top mass. The problem in this unfolding is a strong bias from the top mass used to generate the training data. To compensate this bias we added a global estimate of the top mass to the representation of the measured data and weakened the training bias by including a range of top masses there. As a result of these two structural modifications, the top mass bias was essentially removed.

Using this setup we showed how to extract the top mass along the lines of a recent CMS

analysis [34]. We included two covariance matrices, one describing all statistical uncertainties and one covering the model uncertainty from the training data. We found that, indeed, the impact of the model uncertainty is becoming irrelevant, and that the error in the top mass can be reduced when using the kind of fine binning allowed by the unbinned unfolding method.

Finally, we unfolded the full, 12-dimensional phase space for a given top mass. One failure mode in reproducing the angular distributions was induced by our phase space parametrization. However, a simple lower cutoff on the azimuthal angular separations of the top decay jets allowed for an excellent reproduction of all correlations.

This study serves as a blueprint for an actual CMS analysis, both, for a top mass measurement and for a wider use of the unfolded data. Results for full CMS simulations cannot be shown in this publications, but are available from the CMS members on the author team. Their performance is slightly better than for the fast simulation shown here.

# Acknowledgements

Most importantly, we would like to thank the organizers and experts at the 2024 Terascale Statistics School for pointing out that nobody in their right mind would ever attempt to use unfolding for a mass measurement. We completely agree with that highly motivating point of view.

Moreover, we like to thank Henning Bahl, Anja Butter, Theo Heimel, Nathan Huetsch and Nikita Schmal for many valuable discussions, and Andrea Giammanco and Anna Benecke for useful discussions on the Delphes detector simulation. This research is supported through the KISS consortium (05D2022) funded by the German Federal Ministry of Education and Research BMBF in the ErUM-Data action plan, by the Deutsche Forschungsgemeinschaft (DFG, German Research Foundation) under grant 396021762 – TRR 257: *Particle Physics Phenomenology after the Higgs Discovery*, and through Germany's Excellence Strategy EXC 2181/1 – 390900948 (the *Heidelberg STRUCTURES Excellence Cluster*). We would also like to thank the Baden-Württemberg Stiftung for financing through the program *Internationale Spitzenforschung*, project *Uncertainties – Teaching AI its Limits* (BWST_ISF2020-010). LF is supported by the Fonds de la Recherche Scientifique - FNRS under Grant No. 4.4503.16. SPS is supported by the BMBF Junior Group Generative Precision Networks for Particle Physics (DLR 01IS22079). The research work of DS has been funded by the Austrian Science Fund (FWF, grant P33771). The authors acknowledge support by the state of Baden-Württemberg through bwHPC and the German Research Foundation (DFG) through grant no INST 39/963-1 FUGG (bwForCluster NEMO).

# A   Bias removal methods

As stated in Sec. 3.2, we rely on both, batch-wise conditioning and data augmentation, to unfold the triple jet mass without bias. In Fig. 15 this is demonstrated by showing unfolding results, where we train either without augmentation or without batch-wise conditioning. For both setups, we observe a clear drop in performance when compared to Fig. 8, although all results are produced with the same hyperparameters of Tab. 2. Iterative generative unfolding can ensure prior independence [18], but does not succeed for the triple jet mass in our application. For iterative generative unfolding [18] the first step consists of a generative network to learn the posterior distribution of Eq.(7). For the 4-dimensional unfolding scenario, we look

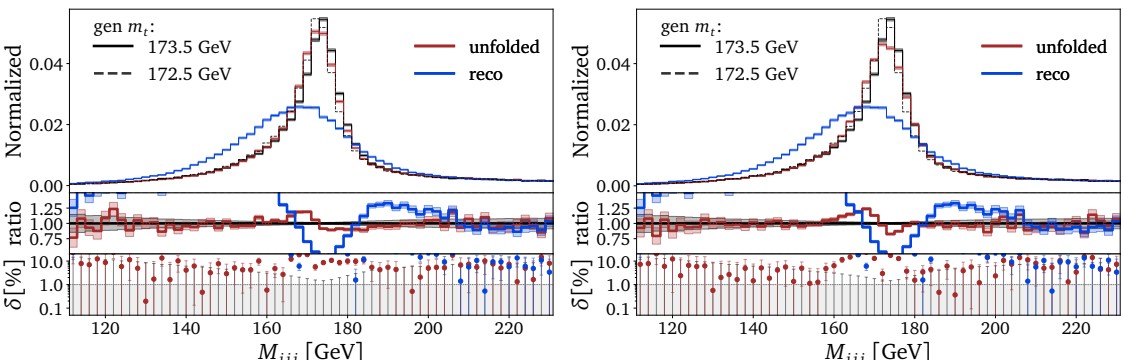

Figure 15: Kinematical distributions from 4-dimensional unfolding. We compare $M_{jjj}$ for $m_t = 172.5\,\text{GeV}$ to generated unfolding for $m_t = 173.5\,\text{GeV}$, not seen during training. On the left panel we show results where the batch-wise condition of Eq.(22) is included into the training pipeline but no augmentation. On the right panel we show results where the training data was augmented with samples of different top masses, but no batch-wise conditioning was included.

at the unfolding results of Fig. 4 as our first step. We train the generative network on MC simulations with $m_t = 172.5\,\text{GeV}$ and try to unfold reco-level pseudo-data with a corresponding top-quark mass of $m_t = 173.5\,\text{GeV}$. In a second step we learn a reweighting between the unfolded pseudo-data and the MC simulation used during training. We see in the lower right panel of Fig. 4 that the unfolded results collapse back to the distribution of the prior MC simulation. The learned reweighting will barely correct the MC simulation, which is confirmed when looking at the learned classifier weights in Fig. 16. They are sharply centered around unity, so we do not gain from the iterations as the MC simulation from the fist iteration matches the MC simulation from the second iteration.

OmniFold [4] learns a classifier-based reweighting between the pseudo-data and the MC simulation on reco-level. In a second step, the OmniFold algorithms pulls the learned reco-level weights to gen-level, event by event, and learns a second classifier-reweighting between the reweighted gen-level distribution and the initial MC gen-level distribution. The procedure can be repeated iteratively. However, for shifted resonances such as the triple jet mass the correction is not learned correctly. This can be confirmed when looking at Fig. 17. Here, we train OmniFold on the 4-dimensional parametrization plus the triple jet mass. The first step correctly reweights the reco-level kinematical distribution of the triple jet mass of the MC

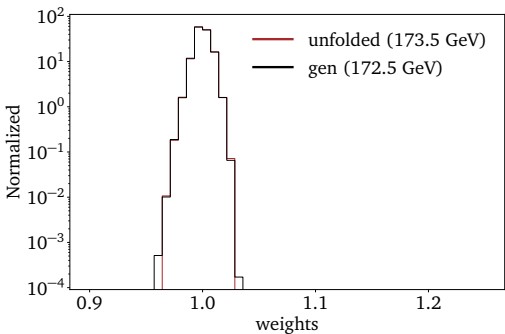

Figure 16: Classifier weights to reweight MC gen-level simulation to unfolded pseudo-data in the 4-dimensional parametrization, as part of the second step in iterative generative unfolding. The top masses are given in parentheses.

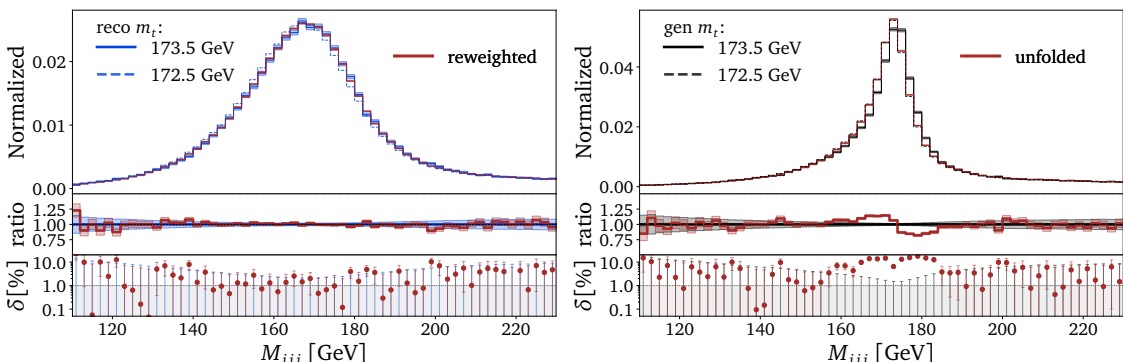

Figure 17: Kinematical distributions from (4+1)-dimensional unfolding. On the left panel, we compare the reco-level $M_{jjj}$ distribution for $m_t = 172.5\,\text{GeV}$ to pseudo-data for $m_t = 173.5\,\text{GeV}$ and the reweighted version computed with the first step of the omnifold algorithm. On the right panel, we make the same comparison on gen-level where unfolded is now the reweighted MC simulation on gen-level.

simulation ($m_t = 172.5\,\text{GeV}$) to our pseudo-data ($m_t = 173.5\,\text{GeV}$). When we pull the learned weights to gen-level, they are not sufficient to reweight the peaked distributions of the gen-level triple jet mass. The reweighted distribution collapses back to the prior MC distribution, indicating again that we cannot remove the prior using iterations. These findings motivate the use of our novel unfolding strategy resulting in Fig. 8.

Although the standard OmniFold approach fails in our unfolding tasks, it does not mean that similar adaptions to the algorithm could not lead to unbiased results. However, we leave a concrete investigation of the matter to the OmniFold authors.

# B   Hyperparameters

| Parameter | |
|---|---|
| LR sched. | cosine |
| Max LR | $10^{-3}$ |
| Optimizer | Adam |
| Batch size | 16384 |
| Network | Resnet |
| Dim embedding | 64 |
| Intermediate dim | 512 |
| Num layers | 8 |

Table 1: Parameters for the 4-dimensional and 6-dimensional networks in Sec. 3.1.

| Parameter | 4D | 6D |
|---|---|---|
| Epochs | 800 | 500(+1000) |
| LR sched. | cosine | cosine |
| Max LR | $10^{-3}$ | $10^{-3}$ |
| Optimizer | Adam | Adam |
| Train batch size | 10000 | 10000 |
| Inference batch size | 50000 | 50000 |
| Dropout | 0.1 | 0.1 |
| Network | Transfusion | Transfusion |
| Dim embedding | 64 | 64 |
| Intermediate dim | 512 | 512 |
| Num layers | 4 | 4 |
| Num heads | 4 | 4 |

Table 2: Parameters for the 4-dimensional and 6-dimensional networks in Sec. 3.2.

| Parameter | 12D |
|---|---|
| Epochs | 500 |
| LR sched. | cosine |
| Max LR | $10^{-3}$ |
| Optimizer | Adam |
| Batch size | 16384 |
| Dropout | 0.1 |
| Network | Transfusion |
| Dim embedding | 128 |
| Intermediate dim | 512 |
| Num layers | 6 |
| Num heads | 4 |

Table 3: Parameters for the 12-dimensional network in Sec. 3.4.

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
