# Peer review of "How to Unfold Top Decays"

_SciPost Physics, doi:SciPost Phys. Core 8, 053 (2025)_

## Round 1 · Referee Report · Anonymous (Referee 1) · 2025-5-23

Strengths

This is a follow up report, based on the second submission. The strengths are unchanged from the first version., though I thank the authors for adding some material that addressed some of my requests.

(1) the authors demonstrate (for the first time) that the hadronic top-decay can be unfolded at the per-event level, allowing the mass of the top decay products to be reconstructed and the top quark mass to be extracted by a fit to a newly-binned distribution.

(2) The authors carry out detailed and robust studies into the unfolding performance, including the underlying ML architecture, methods to reduce biases from the assumed top quark mass in the simulation/training.

(3) The authors compare the performance of their unfolding when extracting the top mass to the results published CMS data

(4) The results demonstrate that the method works as a proof of principle and this opens up future possible applications to LHC data.

Weaknesses

This is a follow up report, based on the second submission. The weaknesses are slightly changed from the first version. in that a few clarifying points are added. Based on the current version, the following weaknesses are present.

(1) The paper does not address the impact of backgrounds, which in particular could bias a constraint on the trijet mass sampled from the 'data' in the training. This is discussed below in the requested changes

(2) Another referee requested additional material comparing to other unbinned methods and the authors added this to the Appendix . It is not clear that the use of Omnifold has been sufficiently optimised to conclude that the authors' method is definitely superior.

Report

I believe that the paper will meet the SciPost requirements for publication once the requested changes have been made. In particular, the paper tries to address the leading theoretical bias in extracting the top mass from unfolded data. It is likely that experimental collaborations will investigate this in the future. The paper therefore sets an agenda that could lead to important breakthrough in the field.

Requested changes

The authors have satisfactorily answered comments (2), (3), (4) and (6) from my original report. I’d like to follow up on the other two comments, (1) and (5), and add an additional remark based on the Omnifold material added to the appendix.

(1) The reply does not answer my question, so I will go into more detail to explain my concern. - To address the first part of your reply: The whole point of adding the M_jjj information to the batch is to strengthen the impact of m_d. Equation 22 specifically states this fact. - The primary message in your paper is that the use of CFM unbinned unfolding can dramatically reduce the leading systematic in the measurement, but only if a measure of the top mass in data is added to the training. - Backgrounds will then impact on the training if they are incorrectly subtracted from the data. This is not equivalent to noise (random fluctuations). - It is crucial to show that the removal of one systematic is not replaced by a different one of equal size. As written currently, the paper proposes a method but does not conclusively show that the new systematics associated with the method are fully controlled. - The way to estimate this is as follows: create an Asimov dataset containing background events at a given normalisation. Apply background subtraction but with a systematically-shifted normalisation, leaving a residual background in the Asimov dataset. Then you can unfold this data and re-extract the top mass to determine the systematic. - You could also test the background subtraction procedure to see if there is a residual bias there, i.e. add a sample of background in and subtract a statistically different sample that has the same normalisation. This is not as critical, but interesting.

(5) Thanks for this confirmation that only paired events are used. I looked at the additions to the text and see that the label ‘paired’ is added to equation 10. I couldn’t find another mention of this. I think a wider explanation is needed here, making the points that: (i) the training is done on paired events and presumably (ii) that you restrict the “data” in the unfolding also to events that satisfy truth&reco [otherwise there would be a bias for events not seen in the training]. The point here is that the efficiencies and fiducial factors are not corrected by this method and additional corrections would be needed. It would be good to add a reference to any literature that shows how that is done.

(7) The addition of Omnifold comparisons (at the request of the other referee) raises the question as to whether this was optimised. For example, I wonder what would happen if the dependence on m_s was also weakened in Omnifold by using the three datasets (this is a standard method in ML training). I welcome the addition of this material, but I think it needs to be more rigorously examined to make sure that claims of superior methodology are fully backed up.

Recommendation

Ask for minor revision

---

## Round 1 · Author Response

Dear Editor-in-Charge,

We addressed all the relevant points raised by the referees and resubmitted the manuscript. We think that this work is highly significant and original as it demonstrates for the first time the inclusion of generative unfolding into an existing LHC analysis pipeline. We clarified our motivations in the updated manuscript. We are not only showing that we can unfold high-dimensional phase spaces to great precision but also reduce the leading systematic uncertainty of the analysis in question with a novel methodology. Therefore, we believe the paper should be considered for publication in SciPost.

Sincerely, The authors

Answer to Report 1

Dear Referee, Many thanks for a thorough consideration of our manuscript. We agree that the next logical step is to include background processes, sideband regions and all systematic uncertainties relevant for the measurement, but such a step goes beyond the scope of this work as this needs to be done within an experiment. We would like to point out that the goal of our paper was to reduce the leading systematic uncertainty, i.e. the choice of m_top in simulations and we successfully achieved this goal introducing a new unfolding strategy. For more details please find our remarks/replies threaded into your report. Sincerely, The authors

(1) Backgrounds: W+jets and single-top are present in the CMS phase space with about 5% contamination from each (see ref [34]). The paper does not address backgrounds and there are some questions that they pose: - Most importantly for the results in this paper: could backgrounds bias the m_jjj^batch requirement in sec 3.2? This is particularly relevant for the W+jets background, which will have a different shape, but might also be true for single top. The paper should explain how this bias would be mitigated during training. Ideally, this bias could be studied in this paper along with any solutions needed to mitigate it. - How can backgrounds be subtracted in this method? I presume this has been studied elsewhere and if so should be cited. If not, some statement needs to be made as to how it can be done.

Since the neural networks are trained on simulations, background events do not affect the training. During inference, the inclusion of background can change the condition M_jjj but the large batch size (50k) and taking the median introduce robustness into the condition. We qualitatively checked that the M_jjj^batch condition is only mildly affected by a low-statistics noise injection with shifted median. We also added a reference to how background subtraction can be done for unbinned unfolding.

(2) Systematics: the paper considers the bias due to the choice of top mass in the training and takes steps to mitigate the bias. However, there are systematic uncertainties that change the shape of the m_jjj distribution. During unfolding, there would presumably be some interplay between these systematics and the steps taken to remove the top mass bias. It isn’t clear whether any impact of these systematics is then smaller, similar to, or larger than the systematics in standard unfolding methods using TUnfold. Systematics that jump to mind include jet energy scale/resolution and the hadronisation/shower models. At minimum, some statements are needed to explain that these systematics exist and qualitatively explain the impact (perhaps by citation to previous work). Ideally, the authors could perform some sort of injection test to show the impact of such systematics on this method.

We included a statement in the text explaining the effects of other systematic uncertainties. They were studied in great detail in the original CMS analysis. In contrast to changes of the top quark mass, variations of the jet energy scale and resolution also affect other observables, such as the reconstructed W boson mass, which was already used in the CMS analysis. Making use of these auxiliary distributions, the effects of jet calibrations and differences in mtop can be disentangled. Therefore, we know that none of those systematic uncertainties lead to an uncontrolled bias in the top mass measurement.

(3) Comparison to CMS data: Fig 10 shows that the statistical precision is much better for the unbinned unfolded method compared to TUnfold and this is stated to be from the finer binning allowed in the analysis. I think the discussion around this needs to be more detailed: - The improvement looks better than simply the finer binning, because the TUnfold stat-only error in fig 10 is +-0.21 whereas the 5-bin CFM fit has a stat-only error of +-0.19 and the CFM-4d 60 bin result has a stat-only error of +-0.17. This feature should be explained in the text. - It is likely that this difference is due to the use of the CMS measurement, which contains background subtraction and also fluctuations in the data itself. Would it not be better to compare apples-to-apples by applying TUnfold directly to your simulated events? - If there is an improvement in stat uncertainty die to finer binning, there might be some tradeoff with worse systematics due to jet energy resolution. This should be discussed. - More trivially, the y-axis range on figure 10 should be reduced as we are most interested in the 0 < Delta Chi^2 < 10.

It is true that we cannot make claims about a reduced uncertainty in the 5-bin CFM fit compared to the TUnfold result. Both uncertainty estimates are very similar (+-0.19 and +-0.21) and are obtained from slightly different setups. Also, the statistical uncertainty in the TUnfold result is already small compared to its systematic uncertainties. We can however show that the CFM unfolding still benefits in terms of statistical precision from more bins. We slightly modified the text in the paper such that this is correctly reported. For this, we include the following sentence: “The statistical uncertainty in the TUnfold result was already small relative to the systematic uncertainties. Both the 4-dimensional and 6-dimensional unfoldings exhibit comparable statistical uncertainties with the 5-bin configuration. However, increasing the number of bins leads to a reduction in statistical uncertainty, as demonstrated below. We also adjusted Fig.10 such that the range on the y-axis is reduced and the x-axis shows the Delta_Chi^2 in the range of +-1 GeV w.r.t to the nominal value of 172.5 GeV, thus containing the masses unseen during training.

(4) Results with and without mjjj sampling: on page 13, it is stated that both the mjjj batch sampling of data and the use of different top masses in the training are required for unbiased results. Could a plot be added to show this? ie showing original bias, inclusion of only mjjj batch sampling, inclusion of only combined training samples, inclusion of both. It would help the reader to understand the relative importance of each step.

Thank you for your suggestion. We included the requested plots to the section “Bias removal methods” in the appendix.

(5) Does the unfolding rely on events being present at both truth and reco, or also correct for truth&!reco and reco&!truth?

Yes, we clarified this point in the text. Our generative unfolding requires paired data.

(6) Text improvements: - finite efficiency -> inefficiency (p4) - recoconstruction -> reconstruction (p5) - section 2.4: this is aiming for a complete description of generstove unfolding, but quite a lot of terms are not defined, ie w(x_gen), w(x_reco), p_latent.

We included all suggested improvements in the text.

Answer to Report 2

Dear Referee,

We are grateful for your feedback. We would like to point out that we rigorously studied other ML-techniques to reduce the bias in the measurement. However, none of those algorithms succeeded in providing an unbiased unfolding result. We welcomed your suggestion and added an appendix with clarifications on the failure of the other methods, with results. We are also emphasizing in the text the originality and the significance of the new unfolding strategy we propose for unbiased unbinned generative unfolding. For more details please find our remarks/replies threaded into your report.

Sincerely, The authors

1) A specific comparison needs to be made against other ML unfolding algorithms - the one paragraph 'explanation' on page 8, whilst true, is still just speculative in this form regardless of "its mathematical foundation". You do not show that others fail or explain why and how they would, nor is this the key message. Simply run the analysis also using at least one other method and stick the results in the appendix or along side the TUnfold methods.

Thank you for this suggestion. We added the section “Bias removal methods” to the appendix, where we study both an iterative generative unfolding approach and an unfolding setup using Omnifold. Both show significant bias in the unfolding of top quark masses that were not used in the training of the models. In fact, the unfolded distributions fell back to the prior completely, such that the updated MC simulations in a second iteration matches the original MC simulation perfectly.

2) If you are making comparisons to TUnfold with respect to biases, the RooUnfold authors documented the accepted statistical procedure in https://arxiv.org/abs/1910.14654 this should be fairly trivial for you to implement and would strengthen the claims of this paper hugely.

The method proposed by the RooUnfold authors cannot be obtained for the CMS result with the data that is publicly available. However, the CMS measurement provides an estimate of the bias due to the “choice of mtop” in the simulation. It is obtained by unfolding a simulated data sample with different top quark mass and comparing the unfolded distribution with its particle level truth. This is repeated for various values of the top quark mass in order to obtain a bias in the unfolding as a function of the true mtop. The “choice of mt” uncertainty is then evaluated at values of +-1GeV compared to mt = 172.5 GeV, which is used in the simulation that is filled into the response matrix. Since we are comparing the CFM method to the existing CMS setup, we chose to obtain a similar measure for the bias.

---

## Round 2 · Author Response

Dear Editor-in-Charge,

We strongly believe scientific progress also follows from discussions. Peer-reviewing is part of this process, where valuable constructive feedback can improve scientific work or clarify the scope of future directions. However, there is a difference between authors and referees - aspects a referee would have liked to follow up if he/she had been an author are not the duty of the actual authors.

Let us make a few specific comments: first, we are developing a method to improve a published CMS analysis, which has been reviewed by the collaboration and by a journal. A critical review and update of this benchmark is not our job, because we have to assume that a published CMS result defines the state of the art. If people feel that the internal CMS review process is insufficient, please contact CMS management. Technically, we do not see how applying the OmniFold reweighting to a range of top mass values is `standard’, please be specific and provide us with a suitable reference. It is also not our goal to develop a new OmniFold strategy to deal with too large weights. As a general comment - of course we would have liked to show how our analysis strategy improves the published CMS analysis in all aspects, but CMS policies explicitly keep us from doing this.

Finally, we cannot but condemn the entirely unprofessional decision of one of the referees to not follow up on the discussions and respectfully ask SciPost to remove this referee from their referee database, so other authors will not have to have the same sobering experience. Sincerely, The Authors

Answer to Report 2

Dear Referee,

Thank you for taking the time to review our manuscript a second time. Our remarks/replies are threaded into your report. Sincerely, The Authors

"(1) The reply does not answer my question, so I will go into more detail to explain my concern. - To address the first part of your reply: The whole point of adding the M_jjj information to the batch is to strengthen the impact of m_d. Equation 22 specifically states this fact. - The primary message in your paper is that the use of CFM unbinned unfolding can dramatically reduce the leading systematic in the measurement, but only if a measure of the top mass in data is added to the training. - Backgrounds will then impact on the training if they are incorrectly subtracted from the data. This is not equivalent to noise (random fluctuations). - It is crucial to show that the removal of one systematic is not replaced by a different one of equal size. As written currently, the paper proposes a method but does not conclusively show that the new systematics associated with the method are fully controlled. - The way to estimate this is as follows: create an Asimov dataset containing background events at a given normalisation. Apply background subtraction but with a systematically-shifted normalisation, leaving a residual background in the Asimov dataset. Then you can unfold this data and re-extract the top mass to determine the systematic. - You could also test the background subtraction procedure to see if there is a residual bias there, i.e. add a sample of background in and subtract a statistically different sample that has the same normalisation. This is not as critical, but interesting."

The effect of the background was studied in the CMS analysis. The uncertainties on the rates were conservatively estimated with 19% for W+jets production, 21% for single top quark production, and 100% for other relevant SM processes. The normalization uncertainties in the different backgrounds introduce a shape uncertainty when changing the normalization of single processes. In the cited CMS paper, the overall background uncertainty was estimated to be only 0.01 GeV in the extraction of the top quark mass and thus negligible compared to other uncertainties. With this in mind, we do not consider the background estimation relevant for this paper, but rather leave the details to the experiments carrying out the full measurement. We believe that there is also a misunderstanding in the formulation of the question. The background samples will never impact the training, which happens only on signal (ttbar) simulation. Background events only enter the analysis when the actual unfolding is done, ie the trained networks are used to solve the inversion problem. After the unfolding, the background would be probabilistically subtracted from the unfolded data. Small shifts in the value of m_d, as would be introduced by background contributions to the M_jjj distribution, will have a negligible impact on the result. Having said that, we would like to point out that the actual amount of background is subject to the experimental analysis. There are many handles for suppressing background if one is willing to pay a price in signal efficiency. The balance in the resulting size of uncertainties is a question of optimization, which the experiments have to do when performing the actual data analysis, something that can not be predicted in our study. Finally, If the background is not well modelled by our simulation hence the subtraction is incorrect, bin-wise subtraction as well as continuous subtraction methods would suffer in the same way.

"(5) Thanks for this confirmation that only paired events are used. I looked at the additions to the text and see that the label ‘paired’ is added to equation 10. I couldn’t find another mention of this. I think a wider explanation is needed here, making the points that: (i) the training is done on paired events and presumably (ii) that you restrict the “data” in the unfolding also to events that satisfy truth&reco [otherwise there would be a bias for events not seen in the training]. The point here is that the efficiencies and fiducial factors are not corrected by this method and additional corrections would be needed. It would be good to add a reference to any literature that shows how that is done."

In all unfolding techniques, one needs to correct for the specific efficiencies of the truth and reco selections. The optimal estimation procedure and size of these efficiencies depend strongly on the details of the event selection and reconstruction. We are aware that non-paired events need to be addressed in a realistic measurement, but we leave the details of the implementation to the experiments. This issue is closely related to the treatment of backgrounds, where non-paired events can be treated as background if they are selected at the reconstruction level but are not part of the measurement’s fiducial phase space at the particle level. On the other hand, events that were generated in the fiducial phase space but were not reconstructed because of the detector’s acceptance or an inefficiency will need to be accounted for by an efficiency correction. There are different ways of dealing with these effects. We added a clarification in the text and also give references for efficiency effects, one of these studies corrects for efficiency effects through a classifier. There, we also write explicitly that only “paired events” are used in our study.

"(7) The addition of Omnifold comparisons (at the request of the other referee) raises the question as to whether this was optimised. For example, I wonder what would happen if the dependence on m_s was also weakened in Omnifold by using the three datasets (this is a standard method in ML training). I welcome the addition of this material, but I think it needs to be more rigorously examined to make sure that claims of superior methodology are fully backed up."

The OmniFold results shown in Fig.17 strictly follow the procedure presented in the original formulation. We consider the optimisation of the training procedure adequate since the first reweighting at detector level correctly shifts, within statistical errors, the Mjjj distribution. The pushed particle-level weights from the second OmniFold step would again bias the detector-level Mjjj distribution, thus never reaching convergence. While we do not exclude the possibility that OmniFold can be extended to the unfolding problem studied in this manuscript, we do not see a clear path towards that extension. Both classifiers trained at step one and two directly compare to the data and they will not benefit from a larger set of simulated events, regardless of the additional conditioning. This study is rather a future development direction for OmniFold than an optimisation issue, which we leave to the Omnifold’s authors.

---

## Round 2 · List of Changes

1. We have added the following discussion the backgrounds at the end of Section 2.1:

“The CMS analysis [34] shows that continuum backgrounds, like $W$+jets production, can be subtracted bin-wise to the level where they are no longer relevant for in the analysis. The normalization uncertainties in the different backgrounds introduce a shape uncertainty when changing the normalization of single processes. While the background normalizations vary between 20-100% in the CMS analysis, the overall background uncertainty was estimated to be only 0.01 GeV in the extraction of the top quark mass and is thus negligible compared to other uncertainties. The method of bin-wise background subtraction can be generalized to the unbinned case with the help of a classifier [52], which suggests that background uncertainties will remain small compared to other systematic uncertainties in this measurement. Therefore, we neglect these in our study and consider signal events only. ”

  1. We have added the following discussion at the end of Section 2.1:

“We only consider paired events in our signal, i.e. events that passed both reco- and gen-level cuts. Non-paired events can be treated as background if they are selected at the reco-level but are not part of the measurement’s fiducial phase space at the gen-level. On the other hand, events that were generated in the fiducial phase space at gen-level but were not reconstructed because of the detector’s acceptance or an inefficiency will need to be accounted for by an efficiency correction. This can be done through weights, as for example done in the Iterative Bayesian unfolding method [42–45] as implemented in RooUnfold [46] and in TUnfold [47], and successfully applied in several jet substructure analyses at the LHC, see for example Refs. [34,48–50]. Another way to include efficiency and acceptance effects is through a classifier [51], but we leave the details of such a study to future work, as these are closely related to the actual implementation of the data analysis.”

  1. We have added the following discussion at the end of Appendix A:

"Although the standard OmniFold approach fails in our unfolding tasks, it does not mean that similar adaptions to the algorithm could not lead to unbiased results. However, we leave a concrete investigation of the matter to the OmniFold authors."

---

## Editorial Decision

published